# SPECTRAL HIGHWAYS: INJECTING HOMOPHILY INTO HETEROPHILIC GRAPHS

## ABSTRACT

It is widely assumed that standard GNNs perform better on graphs with high homophily, leading to the development of specialised algorithms for heterophilic datasets in recent years. In this work, we both challenge and leverage this assumption. Rather than creating new algorithms, we emphasise the importance of understanding and enriching the data. We introduce a novel data engineering technique, *Spectral Highways*, that enhances the performance of both heterophilic and non-heterophilic GNNs on heterophilic datasets. Our method augments a given heterophilic graph by adding supernodes, thereby creating a network of highways connecting spectral clusters in the graph. It facilitates additional paths to bring similar nodes closer than dissimilar ones by reducing the average shortest path lengths. We draw both intuitive and empirical connections between the relative decreases in intraclass and interclass average shortest path lengths and shifts in the graph's homophily levels, providing a novel perspective that extends beyond traditional homophily measures. We conduct extensive experiments on seven heterophilic datasets using various GNN architectures and also compare with data-centric techniques, demonstrating significant improvements in node classification performance. Furthermore, our empirical findings highlight the strong sensitivity of several recent GNNs to the random seed used for data splitting, underscoring the importance of this often-overlooked factor in GNN evaluation.

## 1 INTRODUCTION

In general, real-world networks fall into either of the two categories, i.e. homophilic or heterophilic, decided by a network property called homophily. Homophily is the tendency to connect similar nodes via edge linkage, where class labels of the connected nodes generally govern the notion of similarity. For example, in citation networks, researchers often tend to cite research articles from the same domain (Ciotti et al., 2016). In contrast, low homophily, i.e., heterophily, is observed in heterophilic datasets, where edge formations do not favour similar class labels or actually favour dissimilar class labels. E.g. in social media platforms, people tend to form connections irrespective of gender, whereas, in dating networks, most people prefer to form connections with the opposite gender (Zhu et al., 2021).

A large number of GNN algorithms tend to perform better on homophilic graphs (Xu et al., 2018; Gasteiger et al., 2019; Wu et al., 2019; Deng et al., 2020; Bojchevski et al., 2020; Huang et al., 2021; Brody et al., 2022) and are assumed to be not suitable for graphs with heterophily (Zhu et al., 2020; 2021; Wang et al., 2022; He et al., 2022). This assumption has led to the designing of specialised algorithms for heterophilic datasets. In the recent years, various algorithms have been proposed specifically for heterophilic datasets (Jin et al., 2021; Chen et al., 2020; Chien et al., 2021; Zhu et al., 2020; Lim et al., 2021; Bodnar et al., 2022; Li et al., 2022; Zheng et al., 2022). As highlighted by Platonov et al. (2023), these recently proposed heterophilic GNNs are evaluated on six heterophilic datasets used by Pei et al. (2020) wherein two datasets have a major drawback of train-test data leakage due to the presence of duplicate nodes. Recently Lim et al. (2021) released several new large-scale and diverse heterophilic datasets.

The research for specialised GNNs for heterophilic graphs has been driven by three primary factors (i) the assumption that most GNNs perform better on homophilic graphs (as discussed above), (ii) in heterophilic networks, vertices with high structural and label similarities are likely far away from

each other (Liu et al., 2024; Suresh et al., 2021), and (iii) uniform neighbourhood aggregation and updation is oblivious to the information between similar and dissimilar neighbours (Xu et al., 2023). As discussed in Section 2.2 and Section 5, many specialised methods have attended to the above factors. This motivates us to make further advancement along these directions. In this work, we leverage and challenge the above assumption by injecting homophily into heterophilic graphs, also shown empirically in Section 6. Intuitively, we enable information flow between different regions of the heterophilic graph by comparatively bringing vertices with similar labels closer to each other than the dissimilar ones.

To this end, we make the following **contributions**:

1. We propose *Spectral Highways*, a novel technique that enriches a given heterophilic graph dataset with additional nodes and connections forming highways over the original graph. These highways enable better information exchange between different spectral regions of the heterophilic graph, boosting the performance of both heterophilic and non-heterophilic GNNs for node classification.

2. We empirically relate the performance of Spectral Highways with homophily across heterophilic and homophilic datasets.

3. Empirical findings of our exhaustive experimentation shed light on the high sensitivity of several recently proposed GNNs to the random seed used for data splitting.

4. To the best of our knowledge, we are the first to relate GNN performance with intraclass and interclass average shortest path lengths in graph.

5. We intuitively discuss and empirically show a generic correlation between the changes in graph homophily levels with the relative drop in intraclass and interclass average shortest path lengths.

## 2 RELATED WORK

### 2.1 GRAPH DATASETS

**Homophilic datasets** Preliminary research works in GRL mainly evaluated their algorithms on datasets that possess high homophily. The most widely used datasets for benchmarking are three citation networks, namely Citeseer, Cora and Pubmed (Giles et al., 1998; Sen et al., 2008; McCallum et al., 2000; Namata et al., 2012; Yang et al., 2016), and two co-purchasing networks, namely amazon-photo and amazon-computers (Shchur et al., 2018). Other homophilic datasets used for node classification are citation co-author networks: coauthor-cs and co-author-physics from (Shchur et al., 2018). To evaluate GNNs on large-scale datasets, Hu et al. (2020) created Open Graph Benchmark and introduced highly homophilic datasets for node classification: ogbn-products, ogbn-arxiv, ogbn-proteins, ogbn-mag and ogbn-papers100M.

**Heterophilic datasets** Pei et al. (2020) introduced six graph datasets possessing high heterophily that prompted the designing of specific methods for heterophilic graphs. These six graphs, namely Squirrel, Chameleon, Actor, Texas, Wisconsin, and Cornell, have become the standard benchmarks for evaluating heterophilic GNNs. Platonov et al. (2023) corrected the node duplication in Squirrel and Chameleon datasets and introduced Squirrel Filtered and Chameleon Filtered datasets along with five new medium-size datasets: roman-empire, amazon-ratings, minesweeper, tolokers, and questions. Lim et al. (2021) released seven new large-scale heterophilic datasets, namely Penn94, pokec, arXiv-Year, snap-patents, genius, twitch-gamers, and wiki.

### 2.2 GRL ALGORITHMS

**General GNNs** GNNs have shown their effectiveness on a wide variety of graph learning tasks on real-world datasets. The majority of GNN algorithms are based on the convolution principle which is defined as neighbourhood aggregation and updation. GCN (Kipf & Welling, 2017) aggregates the features of a node's neighbours by learning a weight matrix and uses them to update the node's feature vector. GraphSAGE (Hamilton et al., 2017) samples nodes from the 1-hop and 2-hop neighbourhood for aggregation. GAT (Veličković et al., 2018) uses an attention mechanism

to give varied importance to various neighbours. Xu et al. (2018) introduced Jumping Knowledge networks to capture varied neighbourhood ranges for different nodes where subgraphs have diverse local structures. Wu et al. (2019) proposed a Simple Graph Convolution by successively dropping non-linearities and collapsing weight matrices between consecutive network layers, resulting in a linear classifier following a low pass filter. Gasteiger et al. (2019) explored the relationship between personalised PageRank and GCN to fast approximate the propagation of neural predictions. Liu et al. (2020) proposed DAGNN to decouple representation transformation and propagation in convolution operations. He et al. (2021) introduced BernNet to learn arbitrary graph spectral filters by an order-K Bernstein polynomial approximation. Brody et al. (2022) designed GATv2 to introduce dynamic attention by reversing the order of attention and non-linearity operations in GAT. Topping et al. (2022) studied bottleneck and over-squashing phenomena in message passing neural networks from a geometric perspective. Wang et al. (2023) proposed Allen-Cahn message passing, using interacting particle dynamics, where nodes are particles and edges represent attractive and repulsive forces between particles. Yang et al. (2023) introduced PMLP, which is identical to standard MLP in training but then adopts GNN's architecture in testing. AeroGNN (Lee et al., 2023) highlights vulnerability to over-smoothed features and smooth cumulative attention in attention-based GNNs. Bo et al. (2023) devised Specformer to encode the set of all eigenvalues and performs self-attention in the spectral domain, leading to a learnable set-to-set spectral filter. Huang et al. (2024) proposed UniFilter that integrated the heterophily basis with the homophily basis to construct a universal polynomial basis thus limiting over-smoothing and alleviating over-squashing.

**Heterophilic GNNs**  Pei et al. (2020) directed focus towards heterophilic datasets by introducing Geom-GCN that does bi-level aggregation over the structural neighbourhood obtained by mapping the original graph into a latent continuous space. Zhu et al. (2020) discussed the limitations of GNNs for learning under heterophily and proposed H2GCN. Zhu et al. (2021) proposed CPGNN to learn a class compatibility matrix to model graph homophily. Chien et al. (2021) proposed the use of Generalised PageRank (GPR) for GNN where GPR weights automatically learn to adjust weights in accordance with node label pattern. Lim et al. (2021) proposed LINKX, a simple technique of embedding adjacency matrix and node features separately through MLPs and combining them by concatenation. Fu et al. (2022) introduced $p$-Laplacian based GNN as an approximation of a polynomial graph filter over the spectral domain of $p$-Laplacians. Wang et al. (2022) suggested an adaptive propagation mechanism and aggregation process as per the homophily between node pairs based on attribute and topological information. Li et al. (2022) suggested two models, GloGNN and GLoGNN++, that capture node correlations by learning a coefficient matrix to guide the neighbourhood aggregation further. Maurya et al. (2022) designed FSGNN highlighting the use of softmax as a regulariser and soft-selector of neighbourhood features. Bodnar et al. (2022) proposed neural sheaf diffusion models to achieve linear discrimination of classes in the infinite time limit. GBK_GNN (Du et al., 2022) suggested the use of bi-kernel feature transformation to capture homophily and heterophily followed by a selection gate over kernels for given node pairs. He et al. (2022) suggested block-guided classified aggregation to learn separate aggregation rules for neighbours of varied classes. Luan et al. (2022) proposed Adaptive Channel Mixing to adaptively exploit aggregation, diversification and identity channels node-wisely to extract richer localised information for diverse node heterophily situations. Cavallo et al. (2023) proposed incorporating a learnable importance coefficient per layer to balance the contributions of the neighbourhood and the ego node. Zheng et al. (2023) proposed neural architecture search to build heterophilic GNN models automatically. Liao et al. (2023) decoupled the full-graph dependency from the iterative training and adopted an efficient precomputation algorithm for approximating multi-channel embeddings. Further, we discuss the recent methods that align with our direction of work in Section 5.

## 3 PROPOSED TECHNIQUE

### 3.1 SPECTRAL HIGHWAYS

Spectral Highways (as shown in Fig.1) is a network of highways that run over the top of regions formed by Spectral Clustering over a graph. Spectral Clustering uses connectivity information between data points to form clusters using eigenvalues and eigenvectors of the data matrix. Let $G = (V, E)$ be an undirected graph with vertex set $V = \{v_1, v_2, \ldots, v_n\}$ and edge set $E$. Let $W = (w_{ij})_{i,j=1,\ldots,n}$ be the weighted adjacency matrix of the graph $G$ where $w_{ij}$ represents the

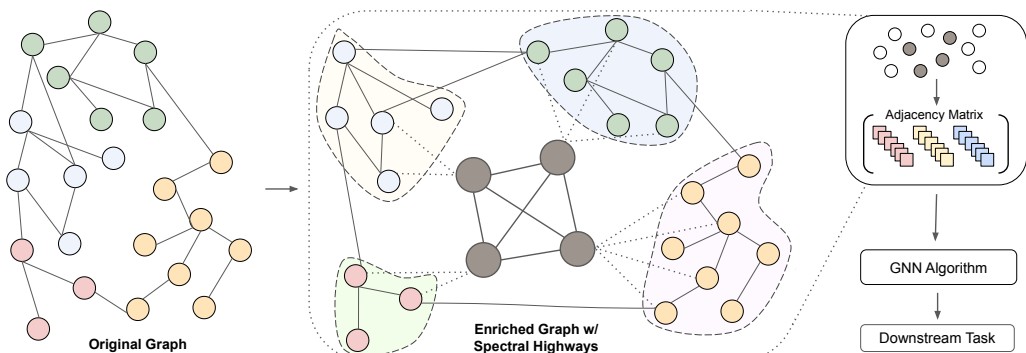

Figure 1: Overview of the use of Spectral Highways. For a given heterophilic graph, we use Spectral Highways to construct an enriched graph. We run available heterophilic or non-heterophilic GNN algorithm on the enriched graph for a downstream node classification task. In this representative enriched graph, the values of $K$, $mincon$ and $pcon$ are 4, 2 and 0.5 respectively. Colour of a node depicts the node belonging to a particular spectral cluster.

edge weight between nodes $v_i$ and $v_j$. If the graph is unweighted, then $w_{ij} = 1$ for an edge present between nodes $v_i$ and $v_j$; otherwise $w_{ij} = 0$. Let $d_i = \sum_{j=1}^{n} w_{ij}$ be the degree of a node $v_i \in V$ and we define degree matrix $D$ as a diagonal matrix with degrees $d_1, \ldots, d_n$ on its diagonal. Then, we can define the unnormalised graph Laplacian matrix as $L = D - W$. We perform Spectral Clustering according to the procedure laid down by Shi & Malik (2000). Let $K$ be the number of clusters we want to construct in $G$. Then, we compute the first $K$ generalised eigenvectors $u_1, \ldots, u_n$ of the generalised eigenproblem $Lu = \lambda Du$. We then stack $u_1, \ldots, u_n$ as column vectors to construct $U \in \mathbb{R}^{n \times K}$. We do not use the popular k-means algorithm (Lloyd, 1982) as it is an iterative scheme sensitive to initialisation, which can lead to poor clusterings. We then directly extract clusters from eigenvectors by cluster_qr method (Damle et al., 2019).

Let $C = \{c_1, \ldots, c_K\}$ be the set of clusters obtained by Spectral Clustering where each such cluster represents a subgraph or a region formed corresponding to the graph topology. We construct highways over the obtained spectral clusters to allow information exchange between different regions of the graph. We instantiate a new node called Spectral node for a cluster $c_i \in C \ \forall \ i \in \{1, \ldots, K\}$. We then connect these Spectral nodes among each other to form a network layer. To construct highways, we need to connect the network of spectral nodes to the underlying graph. For each Spectral node $s_i$, we connect it to the corresponding spectral cluster $c_i$ via a suitable connectivity principle. Instead of making random connections, we define the connectivity principle based on node importance. We propose the use of two popular algorithms to rank the node importance, $rtype$: {PageRank, DivRank}. PageRank (Page, 1999) determines a node's importance by considering the incoming edges it receives from other important nodes in the graph. It outputs a probability distribution over the network to represent the likelihood of a random surfer arriving at a particular node. PageRank relates to the prestige of the nodes in a network, but diversity is another important property that we can account for ranking important nodes. DivRank (Mei et al., 2010) ranks nodes in a network by setting up an interplay between prestige and diversity. Similar to PageRank, DivRank outputs a probability distribution over the network, indicating the node ranking. We experimentally observed both PageRank and DivRank to perform similar in our task.

We then connect a Spectral node $s_i$ to a certain number of nodes in the spectral cluster $c_i$ based on a percentage connectivity parameter $pcon$ consistent across all clusters. We choose percentage as the connectivity measure rather than a fixed integer because spectral clusters are of variable sizes. It ensures that we have a uniform extent of coverage across clusters. For small datasets, we can observe a few clusters that are small in size such that they end up having zero connections as per $pcon$. To account for this scenario, we introduce a $mincon$ parameter that ensures a minimum number of connections to be formed. Still, if the cluster size is too small to accommodate the $mincon$, we do not connect to that cluster and drop the corresponding spectral node.

We have discussed the ranking algorithms and the connectivity coverage above for our connectivity principle. These offer us two new hyperparameter choices, namely $mode$ and $ctype$. We choose

*mode* as a hyperparameter to decide whether to run ranking on a cluster level or graph level, i.e., `local` or `global`. *ctype* decides the type of nodes to choose for making connections. We explore four different ways to select from ranked nodes: `low`, `mid`, `high` and `lmh`. Opting `low` enables connections to the nodes at the bottom of the ranked node spectrum. Similarly, `mid` and `high` lead to connection formation to the nodes in the middle and at the top of the ranked node spectrum, respectively. `lmh` enforces an equally distributed number of link formations with each of the low, mid and high ranked nodes. Intuitively, it may appear to make connections only to the highly important nodes, but empirically, results show no absolute winner for the best choice of *ctype*. Similarly, for *mode*, it may sound better to focus on the local level than the global one, as the Spectral nodes are already connected in a separate network layer to account for global information exchange. However, exhaustive experimentation indicates not to favour any particular *mode* type. Since we design our method to be generic so that a variety of existing GNNs can run across diverse datasets, a one size fits all scenario could not be obtained giving a specific combination of hyperparameters.

The above steps ensure the structural formation of Spectral Highways where nodes (not all) via a highway of Spectral nodes interact with other nodes (not all) in the farther regions in the graph as well as in the same spectral cluster, leading to an enhanced information flow. To initialise the embeddings of a Spectral node, we would not want to compute the average of the representations of nodes forming a connection with it, as this will lead to oversmoothing (Xu et al., 2023). Hence, we initialise the embedding of a spectral node with a random sequence of zeroes and ones keeping the same embedding dimension as those of its neighbouring nodes. To assign a class label to each Spectral node, we take the majority voting of class labels of nodes belonging to the cluster and assign it as the class label of the Spectral node.

Mathematically, we describe Spectral Highways (SH) for a given input graph $G(V, E)$ as a data engineering technique outlined by the following process:

$$SH(G(V, E)) \Rightarrow G'(V', E') \equiv G'(V + S, E + E'' + E''')  \qquad (1)$$

where $S = \{s_1, \ldots, s_K\}$ is the set of Spectral nodes, $E''$ is the set of all possible connections formed amongst the Spectral nodes in the network layer and $E''' = \{N_e(s_1), \ldots, N_e(s_K)\}$.

$N_e(s_i)$ represents the edge neighbourhood of $s_i$ in the underlying graph $G$ and is given by

$$\begin{aligned} N_e(s_i) &= f(mincon, pcon, mode, ctype, rtype, c_i, G) \\ |N_e(s_i)| &= max(mincon, [pcon * |c_i|]_+) \end{aligned} \qquad (2)$$

where $[x]_+$ represents greatest integer less than or equal to $x$. Also, the embedding and the class label of Spectral node $s_i$ is as follows:

$$s_i = [rand\{0, 1\}]^d \; ; \; y(s_i) = M[y(c_i^1), y(c_i^2), \ldots, y(c_i^{|N_e(s_i)|})]  \qquad (3)$$

where $d$ is the dimension of node features, $y$ is the class label, and $M$ is mathematical mode operator. Since, a cluster node $c_i^j$ can belong either to the training set, validation set or testing set, we take $y$ as the true class label only for the training nodes. We assign pseudo labels to validation and testing nodes via modelling a probability distribution $(P)$ over the graph $(G^{tr})$ constituting only the training nodes and the corresponding edges. Let $P_{AB}$ denotes the probability of having an edge between two nodes with class labels A and B in $G^{tr}$ respectively. Hence, for a validation/testing node $c_i^j$, we consider its 1-hop neighbours from the training set denoted by $N_{tr}$. Then the likelihood of assigning a pseudo class label $(l \in L_s)$ is given by $\mathcal{L}(l) = \sum_{n_{tr} \in N_{tr}} P_{ly(n_{tr})}$, where $L_s$ is the set of node class labels in $G$, and thus we assign the pseudo class label with the maximum likelihood, i.e., $\arg\max_{l \in L_s} \mathcal{L}(l)$. Hence, the assigning the class label to Spectral node constitutes three operations: (i) local label profiling captured by summation operator in $\arg\max_{l \in L_s} \mathcal{L}(l)$ (ii) spectral label profiling captured by $M$ (iii) global label profiling encapsulated in $P(G^{tr})$.

## 4  EXPERIMENTS

**Experimental setup**   We conduct extensive experimentation for node classification on a variety of heterophilic datasets using both heterophilic and non-heterophilic GNNs. As Spectral Highways augments the existing heterophilic graph, its merit is determined by the performance of downstream GNNs. We take a heterophilic graph and use Spectral Highways to generate an enriched graph and

then run an available GNN model on this enriched graph to predict the class of a node. For a fair comparison, we only keep all the Spectral nodes in the train set and do not use them for validation or in the test set. We use different GNN hyperparameters for Spectral Highways as the underlying graph is now modified. For each dataset, we consider 5 different random seeds (Appendix A) for data split and run 3 rounds of experiments for each of the splits. Following (Fu et al., 2022), we take 60/20/20 as the train/val/test split ratio. All the experiments are run for 100 epochs. We choose the commonly used accuracy as a metric and report its mean and standard deviation over the 15 runs. We run all experiments on 1 NVIDIA A100 80GB GPU. We share the details of all the hyperparameters used for our models in the supplementary material.

Table 1: Performance comparison of Spectral Highways w.r.t. various models on seven heterophilic datasets. We report the accuracy values for GNN models and Spectral Highways (SH). ChameleonF and SquirrelF represents the filtered versions of Chameleon and Squirrel datasets. arXiv denotes arXiv-Year dataset. We highlight global best result across GNNs for each dataset. Furthermore, we highlight best result for each combination of dataset and GNN. Last column reports the average accuracy jump across datasets observed for a baseline GNN. OOM represents Out Of Memory.

| | Cornell | Texas | Wisconsin | Actor | ChameleonF | SquirrelF | arXiv | Avg (%↑) |
|---|---|---|---|---|---|---|---|---|
| MLP | 84.34 ± 5.86 | 77.00 ± 12.98 | 94.81 ± 5.16 | **43.51 ± 2.72** | 54.13 ± 5.05 | 34.46 ± 10.48 | 39.48 ± 2.26 | |
| SH | **87.17 ± 5.98** | **80.08 ± 6.31** | **95.12 ± 3.31** | 40.81 ± 2.51 | **57.22 ± 3.63** | **44.00 ± 10.18** | **40.11 ± 2.61** | 5.20 |
| GraphSAGE | **86.67 ± 4.10** | 71.83 ± 7.97 | 89.01 ± 5.98 | **40.46 ± 2.26** | 56.94 ± 3.80 | 40.25 ± 8.54 | **50.17 ± 0.60** | |
| SH | 79.29 ± 5.36 | **81.50 ± 6.11** | **93.02 ± 3.32** | 38.67 ± 0.85 | **58.33 ± 2.93** | **46.04 ± 7.64** | 43.74 ± 1.58 | 1.29 |
| GAT | 45.96 ± 14.44 | 56.92 ± 20.98 | 64.94 ± 5.77 | **34.51 ± 1.80** | **58.51 ± 2.74** | **42.65 ± 7.26** | 21.81 ± 4.07 | |
| SH | **47.68 ± 16.13** | **65.75 ± 12.58** | **71.42 ± 5.02** | 33.18 ± 2.11 | 58.44 ± 4.23 | 41.31 ± 2.65 | **41.34 ± 7.72** | 15.95 |
| APPNP | 86.06 ± 6.12 | **81.83 ± 5.09** | 96.60 ± 1.47 | **43.56 ± 3.70** | 59.93 ± 2.64 | 38.53 ± 4.18 | 37.46 ± 6.24 | |
| SH | **86.67 ± 7.13** | 80.50 ± 5.86 | **96.98 ± 2.58** | 41.47 ± 1.91 | **61.94 ± 2.16** | **42.20 ± 10.46** | **39.61 ± 2.72** | 1.89 |
| GPRGNN | 82.02 ± 9.93 | 75.75 ± 12.29 | 92.96 ± 3.18 | **41.80 ± 2.09** | 60.52 ± 2.94 | **45.91 ± 3.90** | 21.58 ± 6.76 | |
| SH | **82.73 ± 5.08** | **78.75 ± 7.92** | **94.14 ± 4.19** | 39.53 ± 1.85 | **60.97 ± 2.07** | 38.90 ± 8.43 | **37.95 ± 9.00** | 8.85 |
| LINKX | 67.88 ± 14.22 | 62.42 ± 14.60 | 81.17 ± 9.27 | 33.88 ± 3.55 | 57.74 ± 2.98 | 43.14 ± 8.33 | **52.94 ± 2.43** | |
| SH | **81.82 ± 7.58** | **78.67 ± 11.51** | **95.12 ± 3.00** | **35.62 ± 3.80** | **61.46 ± 3.91** | **47.44 ± 6.29** | 45.40 ± 4.16 | 10.15 |
| GATv2 | 39.49 ± 22.88 | 48.67 ± 28.00 | 65.06 ± 8.24 | **33.27 ± 1.87** | 57.60 ± 2.98 | 42.56 ± 6.15 | 24.86 ± 7.94 | |
| SH | **48.38 ± 14.73** | **61.25 ± 9.89** | **73.21 ± 3.51** | 32.10 ± 2.06 | **58.89 ± 3.99** | **43.53 ± 3.95** | **44.84 ± 3.60** | 20.32 |
| pGNN | 73.03 ± 10.41 | 68.83 ± 8.58 | 80.06 ± 6.87 | 33.79 ± 2.18 | 58.19 ± 3.84 | **48.90 ± 3.58** | 41.11 ± 0.75 | |
| SH | **78.28 ± 9.73** | **81.00 ± 6.88** | **85.99 ± 3.62** | **35.15 ± 1.90** | **58.92 ± 3.37** | 46.05 ± 4.00 | **42.26 ± 1.32** | 4.93 |
| DAGNN | 60.30 ± 14.15 | 55.00 ± 21.66 | 71.98 ± 4.78 | 34.10 ± 2.44 | 59.34 ± 3.26 | **39.18 ± 5.87** | 23.21 ± 9.28 | |
| SH | **62.42 ± 7.27** | **68.17 ± 12.57** | **72.72 ± 3.05** | **34.36 ± 2.62** | **59.51 ± 3.12** | 37.21 ± 8.84 | **40.70 ± 9.39** | 14.26 |
| BernNet | 83.74 ± 4.91 | 82.67 ± 4.35 | 93.52 ± 3.25 | **38.80 ± 1.32** | 58.89 ± 2.20 | 42.83 ± 3.08 | 22.88 ± 2.78 | |
| SH | **86.77 ± 3.64** | **83.83 ± 6.17** | **97.04 ± 2.13** | 37.63 ± 1.44 | **61.74 ± 2.97** | **43.33 ± 7.16** | **47.09 ± 2.22** | 16.79 |
| AeroGNN | **52.12 ± 32.38** | 35.67 ± 34.01 | 58.83 ± 12.26 | 29.38 ± 11.38 | 47.36 ± 9.96 | 39.70 ± 29.09 | **29.92 ± 17.77** | |
| SH | 47.27 ± 25.15 | **42.83 ± 38.09** | **63.40 ± 19.66** | **32.39 ± 5.80** | **48.75 ± 9.28** | **50.14 ± 24.95** | 29.36 ± 18.39 | 8.02 |
| DirSAGE | 71.41 ± 11.01 | 79.00 ± 11.03 | 92.41 ± 4.48 | **38.80 ± 1.91** | 57.15 ± 5.76 | 42.58 ± 7.64 | 42.43 ± 2.21 | |
| SH | **76.97 ± 7.54** | **81.83 ± 11.00** | **93.40 ± 3.73** | 38.30 ± 2.20 | **59.27 ± 2.87** | **52.96 ± 4.40** | **44.45 ± 2.72** | 6.28 |
| PMLPGCN | 41.01 ± 14.58 | 37.83 ± 26.40 | 55.62 ± 11.22 | 30.53 ± 6.36 | **57.78 ± 2.79** | 50.27 ± 5.15 | 34.56 ± 7.52 | |
| SH | **46.87 ± 20.49** | **61.92 ± 15.75** | **68.58 ± 7.76** | **31.90 ± 5.00** | 54.41 ± 2.71 | **56.39 ± 7.44** | **37.42 ± 12.85** | 17.19 |
| PMLPAPPNP | 29.80 ± 17.89 | 24.67 ± 24.66 | 52.84 ± 14.45 | **31.86 ± 7.19** | 57.12 ± 4.38 | 46.66 ± 9.71 | 34.41 ± 10.46 | |
| SH | **48.79 ± 23.37** | **68.83 ± 13.67** | **63.15 ± 11.62** | 31.83 ± 5.15 | 53.58 ± 4.96 | **54.45 ± 8.04** | **34.88 ± 15.19** | 37.70 |
| UniFilter | 27.47 ± 32.83 | 30.25 ± 37.25 | **74.32 ± 26.48** | 32.41 ± 10.30 | 41.81 ± 8.93 | 25.27 ± 16.69 | 22.69 ± 16.35 | |
| SH | **53.64 ± 33.92** | **42.67 ± 42.14** | 61.36 ± 31.48 | **34.72 ± 5.69** | **45.07 ± 9.11** | **31.21 ± 18.21** | **29.33 ± 15.59** | 26.65 |
| Specformer | **55.56 ± 31.16** | 37.00 ± 32.31 | **75.56 ± 21.29** | 30.31 ± 7.58 | 43.40 ± 17.10 | **38.73 ± 24.08** | OOM | 7.43 |
| SH | 55.35 ± 23.93 | **53.08 ± 36.81** | 70.31 ± 24.56 | **35.80 ± 9.03** | **45.59 ± 18.57** | 33.04 ± 16.99 | OOM | |

**Baseline GNNs** We employ various neural architectures as baseline models and compare their respective performances with the use of Spectral Highways. Hence, for exhaustive benchmarking, we choose: **Only node features** (MLP), **General GNNs** (GraphSAGE (Hamilton et al., 2017), GAT (Veličković et al., 2018), APPNP (Gasteiger et al., 2019), GATv2 (Brody et al., 2022), DAGNN (Liu et al., 2020), BernNet (He et al., 2021), AeroGNN (Lee et al., 2023), PMLP (Yang et al., 2023), Specformer (Bo et al., 2023), UniFilter (Huang et al., 2024)) and **Heterophilic GNNs** (GPRGNN (Chien et al., 2021), LINKX (Lim et al., 2021), pGNN Fu et al. (2022), DirSAGE (Rossi et al., 2024)). Further, we consider two versions of PMLP based on GCN and APPNP.

**Benchmark datasets** For benchmarking and evaluating the performance of our proposed technique, we choose seven datasets with varied statistics, as shown in Table 5 (Appendix A). We choose **Cornell**, **Texas**, **Wisconsin**, and **Chameleon Filtered** heterophilic datasets for their small

size; **Squirrel Filtered** and **Actor** datasets for their medium size; and **arXiv-Year** dataset for its large size. We could not take other datasets like pokec, genius, wiki, etc., as their experiments ran out of memory, and twitch-gamers due to resource constraint. Cornell, Texas and Wisconsin are datasets of WebKB [1] page data gathered from computer science departments of various universities. Lim et al. (2021) introduced arXiv-Year dataset with the task of predicting the year of publication or patent grant in citation network. Squirrel and Chameleon datasets are introduced for node prediction by Pei et al. (2020) and have been extensively used for evaluating heterophilic GNNs. Recently, Platonov et al. (2023) identified the issue of node duplication in these datasets and released their corrected versions, namely Squirrel Filtered and Chameleon Filtered.

**Results** Table 1 shows the performance of several models with and without applying Spectral Highways (SH) on various heterophilic datasets. We see average accuracy improvements for a GNN across all datasets ranging from $1.29\% - 37.7\%$ as shown in the last column of Table 1. We observe that the Wisconsin dataset obtains the highest accuracy, whereas the Actor dataset proves to be the toughest to learn. The highest improvement in accuracy averaged over all GNNs is observed for the Texas dataset with a value of $30.06\%$. Also, we achieve the best performance across all models on 5 out of 7 datasets. Interestingly, the experimental results reveal that recently proposed GNNs like AeroGNN, PMLP, UniFilter and Specformer yield very high standard deviations in accuracy, clearly depicting that their performance largely depends on the random seed used for data splitting. Furthermore, we show an ablation study removing the connection between spectral nodes in Appendix B. Also, we analyse the time and space aspect of Spectral highways in Appendix C and Appendix D respectively.

## 5   COMPARISON WITH DATA-CENTRIC/REWIRING TECHNIQUES

At present, two different lines of thought prevail in the GRL field. One set of work discusses the performance of GNNs regardless of the homophily levels (Luan et al., 2023), or the idea of good homophily and bad homophily (Ma et al., 2022), or the heterophily not always being harmful to GNN's performance (Luan et al., 2022). The other set of work shows that GNN's performance is indeed proportional to the homophily (Rossi et al., 2024; Liu et al., 2024; Xu et al., 2023; Suresh et al., 2021). DirGNN (Rossi et al., 2024) showed that treating graphs as directed improves learning on heterophilic graphs and attributed it to the increase in homophily. SIGMA (Liu et al., 2024) used SimRank (Jeh & Widom, 2002) as an aggregator to establish distinct relationships between similar nodes even when they are not connected and bypassing dissimilar nodes in the local neighbourhood. ALT (Xu et al., 2023) presented a data-centric solution by decomposing the original graph into two modified graphs and using a mixture of complementary filters. WRGNN (Suresh et al., 2021) transformed the input graph, keeping the same number of nodes, into a computation graph containing proximity and structural information as distinct types of edges. They showed that this obtained multi-relational graph possessed an enhanced level of assortativity. The above-discussed methods modify the original graph and use an existing GNN for prediction but not from the principle of inserting super nodes like Spectral Highways. Specifically, Spectral Highways is a data augmentation technique, whereas the above techniques are only data-centric. Azabou et al. (2023) introduced HalfHop that upsamples edges in the original graph by adding "slow nodes" at each edge that can mediate communication between a source and a target node. Qian et al. (2024) proposed IPRMPNN which integrate implicit probabilistic graph rewiring into MPNNs to alleviate the under-reaching and over-squashing issues. We then empirically compare our method with the above discussed methods. We also show a comparison with (Luan et al., 2022) that discusses heterophily not always being harmful and proposes Adaptive Channel Mixing (ACM) and a measure called Aggregated Similarity/Homophily.

We consider GCN and GAT variants of WRGNN, APPNP variant of ALT, SAGE for Dir-GNN, and ACMGCN++ and ACMIIGCN++ variants of ACM. We showed the results for Spectral Highways with GNN variants of DirSAGE, LINKX, and BernNet. We could not report the results on the arXiv-Year dataset as it led to out-of-memory (OOM) for many of the compared methods. The results in Table 2 show that Spectral Highways performs best on all datasets except Actor, which is just second to DirSAGE. Interestingly, we also observe that ACM yields high standard deviations in predictive performance, just like AeroGNN, PMLP, UniFilter and Specformer. To reiterate, we

---

[1]http://www.cs.cmu.edu/afs/cs.cmu.edu/project/theo-11/www/wwkb/

separately construct Table 2 to compare different rewiring models. The results in Table 2 shows the baseline GNNs chosen by the authors of the proposed data centric/ rewiring methods. For example, WRGNN choose only GCN and GAT in their proposed work. We do not create any separate artificial setups that do not belong to the original proposed works.

Table 2: Comparison of Spectral Highways with other data-centric/rewiring techniques. We highlight the best result and the second best for each dataset, respectively, clearly showing SH performing significantly better than the compared methods on 5 out of 6 datasets.

|  | Cornell | Texas | Wisconsin | Actor | ChamF | SquiF |
|---|---|---|---|---|---|---|
| WR-GCN | 64.62 ± 6.76 | 77.81 ± 5.01 | 71.73 ± 5.79 | 34.64 ± 0.98 | 41.64 ± 3.30 | 35.06 ± 3.48 |
| WR-GAT | 64.44 ± 7.25 | 78.51 ± 6.12 | 76.53 ± 4.81 | 35.29 ± 1.05 | 40.00 ± 3.07 | 38.06 ± 2.07 |
| ALT-APPNP | 51.83 ± 7.12 | 58.62 ± 4.24 | 63.58 ± 5.30 | 34.13 ± 0.50 | 39.35 ± 2.03 | 35.66 ± 0.99 |
| SIGMA | 64.56 ± 8.07 | 78.07 ± 7.62 | 79.87 ± 6.44 | 32.93 ± 0.90 | 43.71 ± 3.21 | 40.10 ± 2.07 |
| HalfHop | 47.68 ± 24.13 | 34.83 ± 15.40 | 60.19 ± 12.68 | 25.24 ± 5.41 | 39.34 ± 10.64 | 39.17 ± 10.44 |
| IPRMPNN | 72.32 ± 2.37 | 78.26 ± 1.96 | 80.70 ± 0.85 | 36.31 ± 0.60 | 55.23 ± 1.63 | 42.10 ± 2.16 |
| Dir-SAGE | 71.41 ± 11.01 | 79.00 ± 11.03 | 92.41 ± 4.48 | 38.80 ± 1.91 | 57.15 ± 5.76 | 42.58 ± 7.64 |
| ACMGCN++ | 44.75 ± 27.75 | 44.50 ± 39.23 | 66.98 ± 25.33 | 27.98 ± 12.78 | 50.69 ± 13.97 | 32.56 ± 22.59 |
| ACMIIGCN++ | 50.00 ± 30.84 | 41.33 ± 38.87 | 67.47 ± 25.08 | 28.99 ± 12.72 | 47.85 ± 17.23 | 30.10 ± 19.09 |
| SH (DirSAGE) | 76.97 ± 7.54 | **81.83 ± 11.00** | 93.40 ± 3.73 | **38.30 ± 2.20** | 59.27 ± 2.87 | 52.96 ± 4.40 |
| SH (LINKX) | **81.82 ± 7.58** | 78.67 ± 11.51 | **95.12 ± 3.00** | 35.62 ± 3.80 | **61.46 ± 3.91** | **47.44 ± 6.29** |
| SH (BernNet) | 86.77 ± 3.64 | 83.83 ± 6.17 | 97.04 ± 2.13 | 37.63 ± 1.44 | 61.74 ± 2.97 | 43.33 ± 7.16 |

## 6 ANALYSIS AND DISCUSSION

**Homophily Perspective**    As discussed in Sections 4 and 5, Spectral Highways gives superior performance on several heterophilic datasets and downstream GNN models. To evaluate the assumption that most GNNs perform better on graphs with high homophily, we explored several homophily measures that are available in the literature. Let $G = (V, E)$ is a graph with $n$ nodes, and each node $u \in V$ has a class label $y_u \in \{0, 1, \ldots, C - 1\}$, where $C$ is the total number of classes and $C_k$ represents the set of nodes in class $k$. Node homophily (Pei et al., 2020), which computes the ratio of neighbours that have the same class for an ego node and then computes the mean of these ratios across all nodes. Edge homophily (Zhu et al., 2020) is another standard measure for homophily, which is the fraction of edges connecting two nodes with the same class. Lim et al. (2021) showed that these two simple and intuitive homophily measures are sensitive to the number of classes and their balance, and proposed an Improved homophily measure defined as

$$H_{imp} = \frac{1}{C-1} \sum_{k=0}^{C-1} [h_k - \frac{|C_k|}{n}]_+ \tag{4}$$

where $[a]_+ = max(a, 0)$, and $h_k$ is the class-wise homophily metric defined as

$$h_k = \frac{\sum_{u \in C_k} |\{u \in N(v) : y_v = y_u|}{\sum_{u \in C_k} |N(v)|} \tag{5}$$

Platonov et al. (2022) showed that Improved homophily can also lead to unreliable results and thus proposed a new measure, Adjusted homophily, by correcting the number of intra-class edges by their expected value and is thus insensitive to the number of classes and their balance. Adjusted homophily is based on Edge homophily and is computed as

$$H_{adj} = \frac{H_{edge} - \sum_{k=1}^{C} D_k^2/(2|E|)^2}{1 - \sum_{k=1}^{C} D_k^2/(2|E|)^2} \tag{6}$$

where $D_k = \sum_{v:y_v=k} d(v)$, and $d(v)$ represents the degree of a node $v$.

Luan et al. (2022) proposed Aggregated homophily based on post aggregation node similarity. Please refer the source for more details.

***SH on heterophilic graphs:*** We compute all the above-discussed homophily scores on all seven heterophilic datasets before and after using Spectral Highways. From Table 3, we can observe that Spectral Highways consistently increases the Adjusted homophily and Edge homophily scores

Table 3: Homophily analysis for different homophily measures across 7 heterophilic and 5 homophilic datasets. It shows injection of homophily into heterophilic datasets using Spectral Highways. 'G' represents original graph and 'SH' represents augmented graph after Spectral Highways.

|  | Cornell | Texas | Wisc | Actor | ChamF | SquiF | arXiv | Cora | Cite | Comp | Photo | Pubmed |
|---|---|---|---|---|---|---|---|---|---|---|---|---|
| | | | | | | Node Homophily | | | | | | |
| G | 0.1182 | 0.0872 | 0.1706 | 0.2219 | 0.2481 | 0.1961 | **0.2893** | **0.8251** | **0.7062** | **0.7853** | **0.8364** | **0.7924** |
| SH | **0.1485** | **0.1214** | **0.1926** | **0.2251** | **0.2578** | **0.2063** | 0.2872 | 0.7790 | 0.6651 | 0.7782 | 0.8149 | 0.7589 |
| | | | | | | Edge Homophily | | | | | | |
| G | 0.1321 | 0.1118 | 0.2061 | 0.2194 | 0.2403 | 0.2095 | 0.2181 | **0.8099** | **0.7355** | **0.7772** | **0.8272** | **0.8023** |
| SH | **0.1782** | **0.1892** | **0.2508** | **0.2317** | **0.2596** | **0.2115** | **0.2219** | 0.7038 | 0.6248 | 0.7706 | 0.8164 | 0.7539 |
| | | | | | | Adjusted Homophily | | | | | | |
| G | -0.2029 | -0.2260 | -0.1323 | 0.0061 | 0.0347 | 0.0115 | 0.0051 | **0.7710** | **0.6706** | **0.6823** | **0.7850** | **0.6860** |
| SH | **-0.1018** | **-0.0751** | **-0.0012** | **0.0135** | **0.0545** | **0.0137** | **0.0122** | 0.6393 | 0.5284 | 0.6716 | 0.7705 | 0.6056 |
| | | | | | | Improved Homophily | | | | | | |
| G | **0.0499** | 0 | 0.0495 | 0.0074 | 0.0465 | 0.0409 | **0.0671** | **0.7657** | **0.6267** | **0.7001** | **0.7722** | **0.6641** |
| SH | 0.0301 | **0.0313** | **0.1014** | **0.0171** | **0.0611** | **0.0601** | 0.0662 | 0.6687 | 0.5015 | 0.6906 | 0.7610 | 0.5904 |
| | | | | | | Aggregated Homophily | | | | | | |
| G | **0.2077** | **0.0984** | **0.2829** | **0.2362** | **0.3067** | 0.1053 | 0.1251 | **0.4679** | **0.4385** | **0.3873** | 0.2065 | **0.7094** |
| SH | 0.1823 | 0.0820 | 0.0811 | 0.2226 | 0.2468 | **0.1884** | **0.1403** | 0.1365 | 0.1895 | 0.3569 | **0.2582** | 0.3465 |

across all the datasets. We observe an almost similar trend for Node homophily. As shown in Platonov et al. (2022), Improved homophily leads to unreliable results with no clear pattern in the scores. We also observe a similar unclear pattern in Aggregated homophily. Analysing the results from Table 1 and the homophily scores, we can observe that Spectral Highways achieves better results in datasets where it leads to a high increase in homophily scores.

***SH on homophilic graphs:*** To further verify our hypothesis empirically, we conduct another set of experiments on five commonly used homophilic datasets, namely Cora, Citeseer, Photo, Computers, and Pubmed. We show the statistics of these five datasets in Table 6 (Appendix A). We performed a similar experimental setup for homophilic datasets to that used for heterophilic datasets. The node prediction results are shown in Table 7 (Appendix A), and the homophily scores are reported in Table 3. We observe that using Spectral Highways for homophilic graphs leads to a decrease in the homophily level as measured by all five available homophily measures, with a minor exception in the case of Aggregated homophily. The effect of homophily reduction reflects the drop in performance across almost every homophilic dataset and the chosen GNN.

The exhaustive experimentation provides ample empirical evidence that homophily is a desired network property, enabling most GNNs to perform better. We show empirically that *Spectral Highways injects homophily into heterophilic datasets*, thus justifying the title of the paper. Therefore, we both challenge and leverage the common assumption that most GNNs perform better on high homophily datasets by injecting homophily into heterophilic datasets.

**Beyond Homophily** We design Spectral Highways to enable information flow between different regions of the heterophilic graph by bringing vertices with similar labels closer to each other than the dissimilar ones. Spectral Highways will likely facilitate additional paths between a pair of nodes in a given heterophilic/homophilic graph, potentially reducing the shortest distance or keeping it unchanged for the node pair under consideration. Globally, it reduces the average shortest path in the given graph for nodes with the same class labels as well as different class labels. Intuitively, for a heterophilic graph, where the number of direct connections between similar nodes is less than the dissimilar ones, the additional paths are likely to reduce the average shortest path length for similar nodes comparatively more than the dissimilar nodes. Similarly, in the homophilic graph, where the number of direct connections between dissimilar nodes is less than the similar ones, it brings vertices with dissimilar labels closer to each other than the similar ones.

For a given graph $G$, let $d_{ij}$ denote the shortest path length between a pair of nodes $i$ and $j$. If the two nodes are not connected, we consider it one plus the graph's diameter. Following the notations used above in homophily equations, we define intraclass average shortest path length as follows:

$$ASPL_{SC} = \frac{1}{C} \sum_{k=0}^{C-1} \frac{\sum_{i,j} d_{ij}}{\binom{|C_k|}{2}} \quad \forall\, i, j \in C_k;\ i \neq j \tag{7}$$

and interclass average shortest path length as follows:

$$ASPL_{DC} = \frac{1}{\binom{C}{2}} \sum_{\substack{p,q=1 \\ p<q}}^{C} \frac{\sum_{i \in C_p} \sum_{j \in C_q} d_{ij}}{|C_p||C_q|} \tag{8}$$

As intuitively discussed above, we now empirically show in Table 4 the values of $ASPL_{SC}$ and $ASPL_{DC}$, and the corresponding % drops ($\nabla$) after applying Spectral Highways. Analysing Table3 and Table 4 together offer valuable insights. For heterophilic graphs, the increase in homophily levels (Adjusted Homophily and Edge Homophily) directly correlates with the ASPL Drop Ratio, $ADR = \nabla ASPL_{SC}/\nabla ASPL_{DC}$. We observe the highest homophily injection for Texas, where ADR is the highest, and the lowest homophily injection for Squirrel-Filtered, where ADR is the lowest. Further insights into the results show that Texas obtains the highest performance gains after applying Spectral Highways corresponding to its highest ADR. From Table 1, we see that the Actor is the toughest to fit for various GNNs because it has the same intraclass and interclass ASPL. Also, it obtains minimal accuracy gains after Spectral Highways as it observed the same drops in $ASPL_{SC}$ and $ASPL_{DC}$. For homophilic graphs, the decrease in homophily levels directly correlates with the Inverse ASPL Drop Ratio, $Inverse\ ADR = \nabla ASPL_{DC}/\nabla ASPL_{SC}$.

**To summarise,** we say that increasing homophily in a graph is desirable but we would also like to achieve high ASPL Drop Ratio. Essentially the focus should be to bring similar nodes closer than the dissimilar ones for obtaining better GNN performance.

Table 4: Analysis of average shortest path length computed between nodes with same class (SC) and different class (DC) respectively.

|  | Cornell | Texas | Wisc | Actor | ChamF | SquiF | Cora | Citeseer | Comp | Photo | Pubmed |
|---|---|---|---|---|---|---|---|---|---|---|---|
| | | | | | Average Shortest Path Length – Original Graph | | | | | | |
| SC | 3.262 | 39.307 | 3.161 | 4.101 | 3.835 | 3.139 | 387.016 | 1273.644 | 592.844 | 269.256 | 6.315 |
| DC | 3.3 | 3.205 | 3.229 | 4.101 | 3.921 | 3.165 | 416.893 | 1315.283 | 595.756 | 271.303 | 6.6199 |
| | | | | | Average Shortest Path Length – Spectral Highways | | | | | | |
| SC | 3.032 | 2.518 | 2.81 | 3.578 | 3.334 | 2.992 | 233.757 | 980.329 | 326.735 | 206.88 | 3.661 |
| DC | 3.071 | 2.705 | 2.903 | 3.578 | 3.411 | 3.012 | 236.863 | 995.742 | 327.806 | 207.986 | 3.693 |
| | | | | | % ↓ Average Shortest Path Length – Spectral Highways | | | | | | |
| SC | **7.050** | **93.594** | **11.104** | **12.752** | **13.063** | 4.683 | 39.600 | 23.029 | 44.886 | 23.166 | 42.026 |
| DC | 6.939 | 15.600 | 10.096 | **12.752** | 13.006 | **4.834** | **43.183** | **24.294** | **44.976** | **23.338** | **44.213** |

# 7 CONCLUSION

In this paper, we introduce a perspective of data enrichment that enables better performance of heterophilic and non-heterophilic GNN algorithms on heterophilic graphs by injecting homophily. We propose Spectral Highways that enables better information flow in heterophilic graphs by introducing additional paths, thus bringing similar nodes that may be present in faraway regions in the graph closer to each other. We prove the effectiveness of our technique through several experiments and analyses. We offer a fresh perspective of intraclass and interclass average shortest path lengths beyond homophily. Exhaustive experimentation reveals the high sensitivity of many recent GNNs to the random seed used for data splitting. Our work highlights the importance of data enrichment rather than the need to design specialised models.

**Limitations and Future Directions :** Our work highlights the importance of reducing intraclass ASPL more than interclass ASPL after graph augmentation. Computing ASPL is a costly operation that limits the analysis of massive graphs, like arXiv-Year, in our case. We would like to jointly model intraclass ASPL and interclass ASPL with homophily in a single measure for a given graph without any augmentation to assess the difficulty of GNN in fitting the graph.

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

# A  ADDITIONAL DETAILS

Spectral Highways offer the following hyperparameters to tune mainly in the concise range:

- Number of spectral clusters, $K$: $\{30, 40, 50\}$
- Choice of ranking algorithm, $rtype$: $\{$PageRank, DivRank$\}$
- Percentage connectivity, $pcon$: $\{0.3, 0.4, 0.5\}$
- Minimum number of connections, $mincon$: $\{3\}$
- Mode of ranking, $mode$: $\{$local, global$\}$
- Connectivity type, $ctype$: $\{$low, mid, high, lmh$\}$

We use the official code repositories of the authors for implementing GPRGNN [2], pGNN [3], LINKX [4], DAGNN [5], BernNet [6], AeroGNN [7], DirGNN [8], PMLP [9], UniFilter [10], and Specformer [11]. For the rest of the baseline GNNs, we use the respective implementations in PyTorch Geometric provided by pGNN. We utilise scikit-learn (Pedregosa et al., 2011) implementation of Spectral Clustering. We use five different random seeds for data split as (0, 5, 66, 244, 2020).

Table 5: Statistics of chosen heterophilic datasets.

| Type | Dataset | # Nodes | # Edges | # Features | # Classes |
|---|---|---|---|---|---|
| WebKB Webpage | Cornell | 183 | 295 | 1703 | 5 |
| | Texas | 183 | 309 | 1703 | 5 |
| | Wisconsin | 251 | 499 | 1703 | 5 |
| Actor Co-occurrence | Actor | 7,600 | 33,544 | 931 | 5 |
| Wikipedia Webpage | Chameleon filtered | 890 | 8,904 | 2,325 | 5 |
| | Squirrel filtered | 2223 | 47,138 | 2,089 | 5 |
| Citation Network | arXiv-Year | 169,343 | 1,166,243 | 128 | 5 |

Table 6: Statistics of chosen homophilic datasets.

| Dataset | # Nodes | # Edges | # Features | # Classes |
|---|---|---|---|---|
| Cora | 2,708 | 5,278 | 1,433 | 7 |
| Citeseer | 3,327 | 4,552 | 3,703 | 6 |
| Photo | 7,487 | 119,043 | 745 | 8 |
| Computers | 13,381 | 245,778 | 767 | 10 |
| Pubmed | 19,717 | 44,324 | 500 | 3 |

---

[2] https://github.com/jianhao2016/GPRGNN
[3] https://github.com/guoji-fu/pGNNs
[4] https://github.com/CUAI/Non-Homophily-Large-Scale
[5] https://github.com/divelab/DeeperGNN
[6] https://github.com/ivam-he/BernNet
[7] https://github.com/syleeheal/AERO-GNN
[8] https://github.com/emalgorithm/directed-graph-neural-network
[9] https://github.com/chr26195/PMLP
[10] https://github.com/kkhuang81/UniFilter
[11] https://github.com/DSL-Lab/Specformer

Table 7: Performance comparison of Spectral Highways w.r.t. various models on five homophilic datasets. We report accuracy for GNN models and Spectral Highways. Performance drop is observed for all GNNs as expected due to the decrease in homophily.

| | Cora | Pubmed | Citeseer | Computers | Photo |
|---|---|---|---|---|---|
| MLP | 76.64 ± 1.37 | 88.69 ± 0.57 | 77.03 ± 1.23 | 86.47 ± 1.58 | 90.44 ± 2.69 |
| SH | 76.77 ± 1.39 | 88.56 ± 0.53 | 77.23 ± 0.82 | 86.71 ± 1.07 | 91.69 ± 2.09 |
| GraphSAGE | 88.93 ± 1.22 | 90.83 ± 0.47 | 81.44 ± 1.49 | 87.10 ± 1.60 | 92.41 ± 1.91 |
| SH | 84.45 ± 1.72 | 90.26 ± 0.35 | 79.63 ± 1.14 | 86.46 ± 1.21 | 91.60 ± 1.22 |
| GAT | 89.42 ± 0.73 | 90.31 ± 0.30 | 82.12 ± 1.46 | 89.29 ± 0.85 | 93.85 ± 0.59 |
| SH | 84.76 ± 1.22 | 88.60 ± 0.31 | 80.90 ± 1.09 | 86.94 ± 0.89 | 92.21 ± 0.69 |
| APPNP | 88.83 ± 0.65 | 89.25 ± 0.48 | 81.73 ± 1.73 | 88.73 ± 0.77 | 94.48 ± 0.85 |
| SH | 83.82 ± 1.05 | 89.16 ± 0.53 | 80.66 ± 0.68 | 88.15 ± 1.02 | 94.09 ± 1.04 |
| GPRGNN | 89.76 ± 1.00 | 91.56 ± 0.36 | 82.48 ± 1.73 | 88.94 ± 1.18 | 93.26 ± 1.34 |
| SH | 85.77 ± 1.94 | 89.74 ± 0.25 | 81.84 ± 1.06 | 86.42 ± 2.93 | 92.56 ± 1.12 |
| LINKX | 81.22 ± 2.78 | 88.09 ± 0.96 | 74.18 ± 1.27 | 89.50 ± 1.03 | 94.65 ± 1.07 |
| SH | 70.48 ± 5.81 | 87.88 ± 0.70 | 68.82 ± 2.05 | 88.14 ± 0.80 | 94.02 ± 0.92 |
| GATv2 | 88.95 ± 1.05 | 90.34 ± 0.33 | 82.06 ± 0.94 | 90.19 ± 0.59 | 93.90 ± 0.80 |
| SH | 85.40 ± 1.13 | 88.71 ± 0.26 | 81.01 ± 1.40 | 87.43 ± 0.67 | 92.32 ± 0.56 |
| pGNN | 89.94 ± 1.43 | 91.75 ± 0.29 | 81.28 ± 1.10 | 89.30 ± 0.71 | 94.09 ± 0.91 |
| SH | 83.11 ± 1.05 | 90.00 ± 0.52 | 78.30 ± 0.94 | 86.91 ± 1.46 | 92.31 ± 1.32 |
| DAGNN | 89.61 ± 1.16 | 91.97 ± 0.43 | 81.81 ± 1.21 | 87.34 ± 7.13 | 93.07 ± 3.22 |
| SH | 84.92 ± 1.46 | 89.54 ± 0.38 | 80.28 ± 1.32 | 80.15 ± 10.46 | 80.25 ± 2.39 |
| BernNet | 89.52 ± 0.83 | 90.75 ± 0.63 | 82.13 ± 0.91 | 77.96 ± 19.51 | 82.38 ± 31.82 |
| SH | 84.26 ± 2.41 | 90.17 ± 0.37 | 80.54 ± 1.01 | 79.21 ± 8.64 | 89.45 ± 7.18 |
| AeroGNN | 40.51 ± 23.92 | 55.53 ± 6.70 | 33.97 ± 13.89 | 29.13 ± 17.03 | 36.99 ± 17.85 |
| SH | 35.21 ± 22.33 | 56.04 ± 5.04 | 32.90 ± 7.29 | 17.27 ± 18.74 | 21.37 ± 18.70 |
| DirSAGE | 85.14 ± 3.45 | 90.41 ± 0.52 | 79.40 ± 1.50 | 89.65 ± 0.96 | 95.85 ± 0.70 |
| SH | 84.08 ± 1.84 | 90.75 ± 0.38 | 77.53 ± 1.18 | 89.00 ± 1.49 | 95.49 ± 0.72 |
| PMLPGCN | 78.19 ± 9.25 | 89.47 ± 0.54 | 74.52 ± 6.03 | 83.18 ± 3.46 | 81.74 ± 11.59 |
| SH | 72.34 ± 8.10 | 85.68 ± 1.55 | 72.90 ± 3.51 | 82.93 ± 4.30 | 76.26 ± 14.48 |
| PMLPAPPNP | 73.63 ± 13.49 | 88.81 ± 1.31 | 73.44 ± 5.46 | 79.63 ± 6.41 | 77.44 ± 17.49 |
| SH | 70.58 ± 9.50 | 86.00 ± 1.51 | 72.87 ± 3.51 | 80.25 ± 8.39 | 72.63 ± 17.48 |
| UniFilter | 35.32 ± 25.68 | 59.45 ± 14.18 | 33.52 ± 7.70 | 28.73 ± 22.65 | 26.37 ± 21.03 |
| SH | 30.37 ± 24.18 | 60.00 ± 14.20 | 29.19 ± 10.43 | 31.26 ± 24.32 | 27.76 ± 19.08 |
| Specformer | 48.38 ± 24.03 | 60.82 ± 18.14 | 43.49 ± 12.45 | 34.31 ± 20.15 | 34.97 ± 19.64 |
| SH | 43.41 ± 24.15 | 60.29 ± 13.17 | 41.11 ± 10.61 | 35.57 ± 25.01 | 35.73 ± 18.91 |

## B  ABLATION STUDY

To show the importance of various steps involved in the construction of Spectral Highways, we conduct three ablation studies as follows. (1) We switch off Spectral Clustering and randomly connect the super nodes to the original graph. (2) We use random labeling for super nodes instead of our designed probabilistic modeling. (3) We do not form connections amongst super nodes. We show the percentage difference for each of these ablations w.r.t. our complete approach in Table 8. The reported numerical values are the average changes observed in accuracy across all the discussed GNNs as chosen in Table 1.

Table 8: Ablation analysis showing the impact of switching off Spectral Clustering (A1), probabilistic modeling (A2), and connections between spectral nodes (A3) respectively. The table shows the percentage difference for each of these ablations w.r.t. our complete approach. The reported numerical values are the average changes observed in accuracy across all the discussed GNNs.

|    | Cornell | Texas | Wisconsin | Actor | ChameleonF | SquirrelF | arXiv |
|----|---------|-------|-----------|-------|------------|-----------|-------|
| A1 | 30.97   | 35.01 | 13.03     | 3.55  | 5.47       | 11.26     | 5.12  |
| A2 | 5.78    | 17.57 | 6.53      | 5.45  | 6.04       | 6.85      | 2.48  |
| A3 | 7.14    | 13.50 | 3.81      | 1.62  | 4.66       | 9.51      | 2.07  |

## C  TIME ANALYSIS

In this section, we analyse the time aspect of Spectral Highways in two different aspects. (1) Analyse the model runtime on original graph as well as on the augmented graph, and (2) Analyse the construction time of augmented graph which includes the time taken for (i) Spectral Clustering, (ii) running ranking algorithm, (iii) forming connections between spectral nodes and the original graph and also amongst themselves, and (iv) label and feature assignment of spectral nodes utilising likelihood estimation. We show the results for downstream node prediction task in Table 9 for three GNNs to better analyse the variability in the construction time as different hyperparameters lead to different augmented graphs. We then show the results for graph construction time in Table 10 that also highlights the effect of chosen ranking algorithm. As we can see DivRank considerably takes higher runtime than PageRank. As performance of both DivRank and PageRank are equivalent in our experimentation, we thus prefer PageRank for making connections with spectral nodes.

Table 9: Model runtime comparison for downstream node prediction task. The numerical values represent the time in seconds. values 'G' represents original graph and 'SH' represents augmented graph after Spectral Highways.

|         |    | Cornell | Texas | Wisconsin | Actor | ChameleonF | SquirrelF | arXiv |
|---------|----|---------|-------|-----------|-------|------------|-----------|-------|
| LINKX   | G  | 0.921   | 0.908 | 0.914     | 0.961 | 0.915      | 1.059     | 1.612 |
|         | SH | 1.036   | 1.017 | 1.007     | 1.058 | 1.022      | 1.075     | 1.676 |
| BernNet | G  | 2.554   | 2.662 | 2.658     | 2.537 | 2.19       | 2.343     | 6.146 |
|         | SH | 3.048   | 3.126 | 3.018     | 2.691 | 2.37       | 2.546     | 7.224 |
| DirSAGE | G  | 0.902   | 0.915 | 0.967     | 0.95  | 0.997      | 1.613     | 2.746 |
|         | SH | 1.146   | 1.032 | 1.132     | 1.14  | 1.137      | 1.773     | 3.778 |

Table 10: Graph construction time for downstream node prediction task using different GNNs. The numerical values represent the time in seconds. 'P' represents PageRank and 'D' represents DivRank.

|         | Cornell    | Texas      | Wisconsin  | Actor       | ChameleonF | SquirrelF   | arXiv           |
|---------|------------|------------|------------|-------------|------------|-------------|-----------------|
| LINKX   | 0.290 (P)  | 0.352 (P)  | 0.335 (P)  | 44.637 (D)  | 0.735 (P)  | 2.402 (P)   | 13360.130 (D)   |
| BernNet | 0.326 (P)  | 0.292 (P)  | 0.647 (D)  | 4.998 (P)   | 0.699 (P)  | 18.016 (D)  | 22328.283 (D)   |
| DirSAGE | 0.275 (P)  | 0.341 (P)  | 0.643 (D)  | 5.429 (P)   | 0.752 (P)  | 2.256 (P)   | 3832.083 (P)    |

# D  SPACE ANALYSIS

In this section, we analyse the space aspect of Spectral Highways. Following the setting discussed above in Section C, we uncover the number of nodes and edges that are added through construction of Spectral Highways by comparing their count in the original graph and the augmented graph in Table 11.

Table 11: Space analysis for downstream node prediction task using different GNNs. 'V' and 'E' represents the total number of vertices and edges in the graph. 'G' represents the original graph and 'SH' represents augmented graph after Spectral Highways.

|  | Cornell | | Texas | | Wisconsin | | Actor | | ChamF | | SquirrelF | | Arxiv | |
|---|---|---|---|---|---|---|---|---|---|---|---|---|---|---|
|  | V | E | V | E | V | E | V | E | V | E | V | E | V | E |
| G | 183 | 295 | 183 | 309 | 251 | 499 | 7600 | 33544 | 890 | 8904 | 2223 | 47138 | 169343 | 1166243 |
| SH(LINKX) | 204 | 554 | 211 | 764 | 276 | 893 | 7640 | 37306 | 924 | 9911 | 2263 | 49026 | 169373 | 1242898 |
| SH(BernNet) | 213 | 808 | 210 | 727 | 287 | 1206 | 7640 | 36547 | 918 | 9567 | 2253 | 48680 | 169373 | 1242898 |
| SH(DirSAGE) | 204 | 554 | 205 | 610 | 286 | 1170 | 7630 | 36204 | 918 | 9648 | 2253 | 48680 | 169373 | 1242899 |

