# OpenReview forum: "Spectral Highways: Injecting Homophily into Heterophilic Graphs"
_ICLR.cc/2025/Conference — Submitted to ICLR 2025_

### Official Review · Reviewer_dhKQ · 2024-11-03

**Soundness:** 2
**Presentation:** 2
**Contribution:** 2
**Rating:** 3
**Confidence:** 4

**Summary:**

This paper aims to propose a technique for better performance on heterophilic graphs.
The authors introduce additional virtual nodes to the graph that improve the connectivity of the nodes within clusters.
The experiments are conducted on seven heterophilic graphs.

**Strengths:**

1. The idea is simple and works well.

**Weaknesses:**

1. This paper is written poorly.
2. There is not enough justification for the design of the proposed method.
3. There is no ablation study in the experiments.

**Questions:**

1. The introduction is written poorly. There is no paragraph pointing out the existing challenges of node classification on heterophilic graphs. This makes the last paragraph of contribution very abrupt and lacks motivation. I would recommend adding a paragraph pointing out the current challenges.
2. Again, the related works provide little context or motivation with respect to the proposed method. As this is not a survey paper, these paragraphs are not helpful. This section should serve as a motivation for proposing your work.
3. In line 158, in heterophilic networks, vertices with high similarity are usually 2 steps away. “Far away” is an ambiguous description.
4. Figure 1 does not look like a heterophilic graph, as most edges are connected by the nodes with the same label.
5. The connectivity between the spectral nodes seems redundant. It would be helpful to have an ablation study of removing all the edges between the spectral nodes. Moreover, there is no ablation study at all.
6. In Table 2, most baselines have different base GNN models and make the results not comparable. Dir-SAGE and SH (Dir-SAGE) seem to be comparable, but the results are statistically tied.
7. Section 6 should be moved forward to be right after the proposed method section, to justify the design of the proposed method. However, most analyses are indirect to the reason for performance improvement, and do not help understand why the method works. A theoretical analysis would largely improve the completeness of the paper.
8. In Table 7, using SH hurts the performance in the homophilic graphs most times. In real-world, given a graph with limited labels, knowing whether it is homophilic or heterophilic is not an easy problem. This is important but also not discussed in the paper.

---

> ### Author Response · Authors · 2024-11-25
> **Response to Reviewer dhKQ - Part1**
>
> Dear Reviewer dhKQ,
>
> We thank you for appreciating our idea as simple and working very well.
> ***
> **W1**: This paper is written poorly.
> **W2**: There is not enough justification for the design of the proposed method.
>
> **Response to W1 and W2**:  We have revised our paper to accommodate your concerns regarding writing and including justifications. However, we received motivating and encouraging comments from other reviewers and public commentators for our previous version also. Just highlighting for your perusal:
>
> * The method and illustrations in the paper are clear.
> * The paper organizes existing motivations for heterophilic graphs and addresses them in the analysis section.
> * The story is interesting.
> * The idea proposed in the paper is reasonable and improves on the concept of virtual nodes.
> * I thoroughly enjoyed reading this paper smoothly, especially the sensitivity of many of the latest GNNs to random seed, and the uncovering of shortest path length phenomena and its relationship to the various homophily measures and model performance.
> * I am quite fascinated by your approach of employing traditional ML concepts of Spectral clustering for graph node prediction in a heterophilic setting.
> * It is extremely interesting that you play with the notion of path lengths to bring dissimilar vertices closer to inject homophily.
>
> ***
> **W3**: There is no ablation study in the experiments.
>
> **Response**: Thanks for your feedback. We have included the desired ablation study as you asked in Q5. It has indeed improved the quality of the paper. Please have a look at Appendix Section B in the revised manuscript.
>
> ***
> **Q1**: The introduction is written poorly. There is no paragraph pointing out the existing challenges of node classification on heterophilic graphs. This makes the last paragraph of contribution very abrupt and lacks motivation. I would recommend adding a paragraph pointing out the current challenges.
>
> **Q2**: Again, the related works provide little context or motivation with respect to the proposed method. As this is not a survey paper, these paragraphs are not helpful. This section should serve as a motivation for proposing your work.
>
> **Response to Q1 and Q2**: Thanks for pointing this out. We have tried our best to incorporate your feedback and have added a paragraph in the Introduction section. We hope now our revised manuscript addresses your concerns.
>
> ***
> **Q3**: In line 158, in heterophilic networks, vertices with high similarity are usually 2 steps away. “Far away” is an ambiguous description.
>
> **Response**: We could not locate any study that states vertices with high similarity are usually 2 steps away in heterophilic networks. We believe that the definition of a heterophilic graph, in general, defines the immediate neighbourhood diversity of a node, and it is recursive in nature. Therefore, it may not be necessary to have similar vertices 2 hops away. We have duly cited the prominent works that define the notion of being far away in the next line (presently, Line 54 in the revised manuscript).
>
> ***
> **Q4**: Figure 1 does not look like a heterophilic graph, as most edges are connected by the nodes with the same label.
>
> **Response**: Sorry for the confusion in the figure. We have now clarified the role of colours, i.e. the nodes with the same colour belong to the same spectral cluster. Colour does not show a node’s label. The node colouring is done so that the original graph and the augmented graph structure can be compared visually after the construction of Spectral Highways. We have mentioned that the given graph is a heterophilic graph (though labels are not shown).
>
> ***
> **Q5**: The connectivity between the spectral nodes seems redundant. It would be helpful to have an ablation study of removing all the edges between the spectral nodes. Moreover, there is no ablation study at all.
>
> **Response**: Thanks for suggesting to conduct an ablation study for the spectral nodes. We have included the ablation study in Appendix Section-B.
>
> ***
> **Q6**: In Table 2, most baselines have different base GNN models and make the results not comparable. Dir-SAGE and SH (Dir-SAGE) seem to be comparable, but the results are statistically tied.
>
> **Response**: To compare results, we have shown in Table 1 the performance improvement in the base GNN model due to the Spectral Highways. We separately constructed Table 2 to compare different rewiring models. The results in Table 2 show the baseline GNNs chosen by the authors of the proposed data-centric/ rewiring methods. For example, WRGNN chose only GCN and GAT in their proposed work. We do not create any separate artificial setups that do not belong to the original proposed works. We have now clarified this in the paper to avoid the confusion.

---

> ### Author Response · Authors · 2024-11-25
> **Response to Reviewer dhKQ - Part2**
>
> **Q7**: Section 6 should be moved forward to be right after the proposed method section, to justify the design of the proposed method. However, most analyses are indirect to the reason for performance improvement, and do not help understand why the method works. A theoretical analysis would largely improve the completeness of the paper.
>
> **Response**: Thanks for your suggestion. After due consideration, we found that the original flow is lucid and logically coherent. We have now revised the Introduction to highlight the reasons for our proposed method’s performance. We hope that it will improve the readability.
>
> ***
> **Q8**: In Table 7, using SH hurts the performance in the homophilic graphs most times. In real-world, given a graph with limited labels, knowing whether it is homophilic or heterophilic is not an easy problem. This is important but also not discussed in the paper.
>
> **Response**: In fact, our proposed Spectral Highways can also be used to classify a graph into homophilic or heterophilic using the improvement quantum in homophily scores of available labels. If the homophily score improves, then the graph is heterophilic; otherwise, the graph is homophilic. The other way to classify the graphs can be on the basis of GNN models’ performance gains after applying Spectral Highways. If the performance improves considerably, the graph is heterophilic; otherwise, the graph is homophilic.
>
> We appreciate your concern but we also mention dedicated research that deals specifically with heterophilic graphs ([1-5]).
>
> [1] Edge directionality improves learning on heterophilic graphs. Emanuele Rossi, Bertrand Charpentier, Francesco Di Giovanni, Fabrizio Frasca, Stephan G¨unnemann, and Michael M. Bronstein. Proceedings of the Second Learning on Graphs Conference, volume 231 of Proceedings of Machine Learning Research, pp.25:1–25:27. PMLR, 27–30 Nov 2024
>
> [2] Auto-heg: Automated graph neural network on heterophilic graphs. Xin Zheng, Miao Zhang, Chunyang Chen, Qin Zhang, Chuan Zhou, and Shirui Pan. In Proceedings of the ACM Web Conference 2023, WWW ’23
>
> [3] A simple method for representation learning on heterophilous graphs.  International
> Joint Conference on Neural Networks (IJCNN), pp. 1–8, 2023
>
> [4] Haoyu Liu, Ningyi Liao, and Siqiang Luo. Sigma: Similarity-based efficient global aggregation
> for heterophilous graph neural networks, 2024.
>
> [5] Derek Lim, Felix Hohne, Xiuyu Li, Sijia Linda Huang, Vaishnavi Gupta, Omkar Bhalerao, and
> Ser Nam Lim. Large scale learning on non-homophilous graphs: New benchmarks and strong
> simple methods. Advances in Neural Information Processing Systems, 34:20887–20902, 2021

---

> > ### Comment · Reviewer_dhKQ · 2024-11-26
> >
> > I thank the authors for the responses. I have reviewed the responses and increased the score for the presentation. However, there are still issues, including a lack of justification for why the method works and sacrificing performance on homophilic graphs to improve performance on heterophilic graphs. Therefore, I decided to maintain the rating.

---

> > > ### Author Response · Authors · 2024-11-27
> > > **Response to Reviewer dhKQ**
> > >
> > > Dear Reviewer dhKQ,
> > >
> > > Thanks for increasing the presentation score. However, we have addressed/answered **all** the queries that you had. Moreover, we have now shown **two** more ablation studies over and above your request, which adds further merit to our proposed method. We fail to understand what else you need in justification.
> > >
> > > Regarding your statement, “lack of justification for why the method works and sacrificing performance on homophilic graphs to improve performance on heterophilic graphs.”, we have discussed this in detail in the Analysis and Discussion section 6. The crux of the discussion is mentioned in lines 467-476, which resolves your query.
> > >
> > > **Why the method works?** :: Again, to summarise, our proposed method improves the homophily score of heterophilic graphs. This phenomenon is illustrated through extensive experiments in the paper using existing homophily measures and our new proposed measures of average path lengths. Intuitively, for a heterophilic graph, where the number of direct connections between similar nodes is less than the dissimilar ones, the additional paths reduce the average shortest path length for similar nodes comparatively more than the dissimilar nodes, thus increasing the homophily. That is the reason why our proposed method works for most of the GNN models for the node classification task on heterophilic graphs.
> > >
> > > **Sacrificing the performance on homophilic graphs to improve performance on heterophilic graphs**:: We would like to emphasise strongly that our research is on Heterophilic Graphs and not on homophilic graphs. We have mentioned this throughout the manuscript. Most importantly, the title of the paper, “Spectral Highways: Injecting Homophily into Heterophilic Graphs”, completely resonates with the research conducted. Again, to reiterate, there are dedicated research works on heterophilic graphs, as also pointed out to you in our last response. Furthermore, we are not advocating to augment every input graph with Spectral Highways but apply the augmentation only on heterophilic graphs. Experiments on homophilic graphs in the paper are shown only for the purpose of completeness and studying its effects.

---

### Official Review · Reviewer_bR69 · 2024-11-03

**Soundness:** 2
**Presentation:** 2
**Contribution:** 2
**Rating:** 3
**Confidence:** 3

**Summary:**

This paper proposes a new technique called Spectral Highways that is designed to improve the performance of GNNs on heterophilic datasets. Namely, supernodes are added to the graph, each supernode is linked to a particular cluster and supernodes are linked to each other. This allows better information exchange between different regions of the graph.

**Strengths:**

The idea proposed in the paper is reasonable and improves on the concept of virtual nodes.

**Weaknesses:**

- Most of the datasets used in the experiments are small and have certain flows. Namely, Cornell, Texas and Wisconsin are extremely small and are unbalanced (so, accuracy should not be used as a performance metric). Then, Squirrel Filtered and Chameleon Filtered are used by Platonov et al. (2023) to illustrate the flows of the original Squirrel and Chameleon and are not advised for further usage. The new datasets proposed by Platonov et al. (2023) are not used in this paper. Similarly, only one dataset from Lim et al. (2021) is used. The authors write that "We could not take other datasets like pokec, genius, wiki, etc., as their experiments ran out of memory, and twitch-gamers due to resource constraint." Does this apply to all models? For instance, Platonov et al. (2023) use several simple models that show good results on their datasets. The absence of experiments on realistic datasets reduces the reliability of conclusions made in the paper.
- From the description of the proposed approach it is not clear why one would expect it to work on heterophilic datasets. In particular, it is not obvious why it should improve homophily. As I understand, new edges are added independently of whether they are homophilic. Thus, it seems that edges with neutral homophily are added which agrees with the results reported in Table 3: adding neutral edges to heterophilic graphs makes them more homophilous (adjusted homophily, in most cases, becomes closer to zero), and adding neutral edges to homophilic graphs makes them more heterophilous (again, adjusted homophily becomes closer to zero).
- In lines 208-232 the authors discuss several options for their algorithm and write that it is not always the case that the intuitive option works and thus all the options need to be tried on a particular dataset. Thus, the proposed approach is not so well supported by the intuition described in the text.
- Equation (1) is claimed to be a formal mathematical description of the proposed algorithm, but it is not a valid mathematical expression and thus complicates the reading.
- If I am not mistaken, code for reproducing the results is not provided.
- Some of the ideas of the proposed approach are related to the concept of virtual nodes, so I suggest discussing related work on this topic (e.g., Hwang et al., 2022; Qian et al., 2024). For instance, similarly to the proposed approach, Hwang et al. (2022) cluster the initial graph and assign a virtual node to each cluster (however, there are no edges between the virtual nodes).

Hwang, EunJeong, et al. "An analysis of virtual nodes in graph neural networks for link prediction." The First Learning on Graphs Conference. 2022.

Qian, Chendi, et al. "Probabilistic Graph Rewiring via Virtual Nodes." arXiv preprint arXiv:2405.17311, 2024.

Minor comments:
- The range of indices in equation (8) can be fixed: $p$ and $q$ range from 1 to $C$, $p \neq q$ should be inside the formula, the comment about $i$ and $j$ also should not be outside the formula (since these are indices inside the sum).
- Table 1: For better readability, I suggest highlighting the best result not only in cases when the proposed solution outperforms the baseline.
- Table 7: I suggest highlighting the best algorithms in this table too.

Typos:
- L43: "updation" -> "updates"
- L52: "labels. e.g. In"
- L428: Should be \citet here
- L484: "denotes" -> "denote"

**Questions:**

Q1. What is the main intuition on why the method is expected to work well on heterophilic graphs?

---

> ### Author Response · Authors · 2024-11-25
> **Response to Reviewer bR69 - Part 1**
>
> Dear Reviewer bR69,
>
> We thank you for finding our idea reasonable. Below, we would like to resolve your concerns.
> ***
> **W1**: Most of the datasets used in the experiments are small and have certain flows. Namely, Cornell, Texas and Wisconsin are extremely small and are unbalanced (so, accuracy should not be used as a performance metric). Then, Squirrel Filtered and Chameleon Filtered are used by Platonov et al. (2023) to illustrate the flows of the original Squirrel and Chameleon and are not advised for further usage. The new datasets proposed by Platonov et al. (2023) are not used in this paper. Similarly, only one dataset from Lim et al. (2021) is used. The authors write that "We could not take other datasets like pokec, genius, wiki, etc., as their experiments ran out of memory, and twitch-gamers due to resource constraint." Does this apply to all models? For instance, Platonov et al. (2023) use several simple models that show good results on their datasets. The absence of experiments on realistic datasets reduces the reliability of conclusions made in the paper.
>
> **Response**: We would like to clarify that Cornell, Texas, Wisconsin, Squirrel Filtered, Chameleon Filtered and Actor are the de facto datasets used in Heterophilic Graph Learning. As proof of evidence, we would like to refer you to the suggested and prominent paper [1] by **Reviewer e6x4**. Please have a look at page number 6 for brevity. It also clearly mentions that
> is the chosen performance metric. Additionally, we chose one more large dataset, arXiv, for even exhaustive experimentation.
>
> Again, we would like to mention that we have not used any flawed Squirrel and Chameleon datasets. Rather, we used the new datasets proposed by Platonov et al. (2023), i.e. Squirrel Filtered and Chameleon Filtered. We have mentioned this in several places in our manuscript in the text (line number 281, 323) as well as in Tables 1,2,3,4,5,8,9,10, and 11.
>
> We have taken one of the two big datasets, i.e., arXiv, from Lim et al. (2021) for its large size. Other even bigger datasets did not fit into our GPU machine for any GNN model, including the simpler one. The datasets used in Platonov are rather small in size, and the authors of this paper emphasize the use of large enough datasets to provide statistically significant results but small enough to allow for evaluating the target models.
>
> Hence, we would like to emphasise that the datasets used are de facto and hence realistic, and do not reduce the reliability of our work in any capacity.
>
> [1] Azabou, Mehdi, et al. "Half-Hop: A graph upsampling approach for slowing down message passing." International Conference on Machine Learning. PMLR, 2023.
> ***
> **W2**: From the description of the proposed approach it is not clear why one would expect it to work on heterophilic datasets. In particular, it is not obvious why it should improve homophily. As I understand, new edges are added independently of whether they are homophilic. Thus, it seems that edges with neutral homophily are added which agrees with the results reported in Table 3: adding neutral edges to heterophilic graphs makes them more homophilous (adjusted homophily, in most cases, becomes closer to zero), and adding neutral edges to homophilic graphs makes them more heterophilous (again, adjusted homophily becomes closer to zero).
>
> **Response**: Dear reviewer, we have discussed this in detail in the Analysis and Discussion section 6. The crux of the discussion is mentioned in lines 467-476, which resolves your query. When we add (neutral) edges, they get added both between similar nodes and dissimilar nodes.
>
> Intuitively, for a heterophilic graph, where the number of direct connections between similar nodes is less than the dissimilar ones, the additional paths are likely to reduce the average shortest path length for similar nodes comparatively more than the dissimilar nodes, thus increasing the homophily. Similarly, in the homophilic graph, where the number of direct connections between dissimilar nodes is less than the similar ones, it brings vertices with dissimilar labels closer to each other than the similar ones, thus decreasing the homophily.

---

> ### Author Response · Authors · 2024-11-25
> **Response to Reviewer bR69 - Part 2**
>
> **W3**: In lines 208-232 the authors discuss several options for their algorithm and write that it is not always the case that the intuitive option works and thus all the options need to be tried on a particular dataset. Thus, the proposed approach is not so well supported by the intuition described in the text.
>
> **Response**: Again, like any deep learning model that offers a set of hyperparameters, such as dropout, number of layers, number of hidden units, activation function, etc., and the right combination of such hyperparameters dictates the model’s performance. Similarly, our proposed algorithm offers a set of hyperparameters, and we have duly mentioned a narrow range of searches for them in Appendix Section A. Moreover, we uploaded the chosen hyperparameters in the supplementary file during the submission. Also, our empirical results resonate well with the intuition described in the text.
> ***
> **W4**: Equation (1) is claimed to be a formal mathematical description of the proposed algorithm, but it is not a valid mathematical expression and thus complicates the reading.
>
> **Response**: We thank you for pointing out this. We have corrected the typo in the revised paper.
> ***
> **W5**: If I am not mistaken, code for reproducing the results is not provided.
>
> **Response**: Yes, we have not uploaded the code as it is not a requirement. We will release the code upon acceptance. Hence, we request you to kindly not perceive this as a weakness, as it is a general research practice to safeguard our work. Please refer to the policy of the ICLR conference for the guidelines regarding releasing the code.
> ***
> **W6**: Some of the ideas of the proposed approach are related to the concept of virtual nodes, so I suggest discussing related work on this topic (e.g., Hwang et al., 2022; Qian et al., 2024). For instance, similarly to the proposed approach, Hwang et al. (2022) cluster the initial graph and assign a virtual node to each cluster (however, there are no edges between the virtual nodes).
>
> **Response**: We took sincere note of it. The work of Hwang et al. 2022 on link prediction is not in the domain of node prediction. However, the other work you pointed out (Qian et al. 2024) is quite very recent, and we have thus incorporated it into our manuscript as per your suggestion. Please see it in Table 2 and Section 5.
>
>  ***
> **Minor comments and Typos**: Thanks for your valuable feedback. We have taken each of your minor comments diligently and also corrected the typos. You may please see the revised manuscript.
> ***
> **Q1**: What is the main intuition on why the method is expected to work well on heterophilic graphs?
>
> **Response**: Increase in homophily in a heterophilic graph helps in exploiting the locality for solving the task. In the case of GNN models, due to uniform neighbourhood aggregation and updation without differentiating similar and dissimilar neighbours in heterophilic graphs, the node embeddings intermix, resulting in poor performance. On the other hand, increased homophily due to an increase in the number of similar nodes in the neighbourhood would reduce the impact of dissimilar nodes. Similar observations have also been made in [2][3].
>
> Our proposed method improves the homophily score of heterophilic graphs. This phenomenon is illustrated through extensive experiments in the paper using existing homophily measures and our new proposed measures of average path lengths. We have answered this question in our response to your pointed weakness W2. Intuitively, for a heterophilic graph, where the number of direct connections between similar nodes is less than the dissimilar ones, the additional paths are likely to reduce the average shortest path length for similar nodes comparatively more than the dissimilar nodes, thus increasing the homophily. Similarly, in the homophilic graph, where the number of direct connections between dissimilar nodes is less than the similar ones, it brings vertices with dissimilar labels closer to each other than the similar ones, thus decreasing the homophily.
>
> [2] Node classification beyond homophily: Towards a general solution. In Proceedings of the 29th ACM SIGKDD Conference on Knowledge Discovery and Data Mining, KDD ’23
>
> [3] Sigma: Similarity-based efficient global aggregation for heterophilous graph neural networks, arxiv 2024.
>
> ***
> Thanks for your valuable feedback. We hope that we have resolved all of your comments and have incorporated your suggestions.

---

> > ### Author Response · Authors · 2024-11-29
> > **Response to Reviewer bR69 - Part 3**
> >
> > Dear Reviewer bR69,
> >
> > Please find below the results for a **large** heterophilic dataset, Twitch-Gamers from Lim et al. (2021), containing 1,68,114 nodes and 67,97,557 edges.
> >
> > ***
> >
> > | Model | GraphSAGE | GAT | APPNP | LINKX | GATv2 | PGNN | DAGNN | DirSAGE | PMLPGCN | PMLPAPPNP | UniFilter |
> > |-------------|:-------------:|:------:|:-------------:|:-----:|:-------------:|:-----:|:-----:|:-------------:|:-----:|:-------------:|:-----:|
> > | **G**   |59.26 ± 0.63 |52.04 ± 6.84 |59.66 ± 0.57 |62.62 ± 1.62 |50.60 ± 7.12 |60.06 ± 1.08 |58.49 ± 4.16 |60.36 ± 0.76 |48.68 ± 10.79 |50.47± 8.32 |57.89± 5.44|
> > | **SH** |60.54 ± 0.27 |54.95 ± 5.76 |60.77 ± 0.63 |65.51 ± 1.49 |52.18 ± 6.83 |60.99 ± 1.21 |59.64 ± 1.96 |62.55 ± 0.68 |52.78 ± 6.10 |51.74 ± 9.65|59.04 ± 3.69 |

---

> > > ### Comment · Reviewer_bR69 · 2024-11-30
> > >
> > > Thank you for the experiments on a new dataset, I think that it is important for validating the claims of the paper. I have one additional question regarding the results. Could you please explain the difference in performance between your results and the results in Lim et al. (2021)? This applies to the arXiv-year and twitch-gamers datasets. In particular, the results for the LINKX reported by Lim et al. (2021) are 56 for arXiv-year and 66 for twitch-gamers, which is larger than all the values reported for these datasets in the current paper, both with or without SH. I see similar differences for some other models, so I wonder whether it is explained by differences in data preprocessing (e.g. splits) or training process.

---

> > ### Comment · Reviewer_bR69 · 2024-11-30
> >
> > I thank the authors for their detailed response and updated paper!
> >
> > > We would like to clarify that Cornell, Texas, Wisconsin, Squirrel Filtered, Chameleon Filtered and Actor are the de facto datasets used in Heterophilic Graph Learning. As proof of evidence, we would like to refer you to the suggested and prominent paper [1] by Reviewer e6x4.
> >
> > In general, I do not think that the fact that many papers use some potentially flawed datasets justifies their usefulness. For instance, the original Squirrel and Chameleon datasets are also often used in the papers, while they have significant train-test leakage. The same applies to relying on accuracy for imbalanced datasets.
> >
> > > Again, we would like to mention that we have not used any flawed Squirrel and Chameleon datasets. Rather, we used the new datasets proposed by Platonov et al. (2023), i.e. Squirrel Filtered and Chameleon Filtered.
> >
> > Yes, I understand that the filtered versions are used, but, as I understand, Platonov et al. do not guarantee that the filtration produces meaningful datasets and thus these datasets are not included in the list of five datasets proposed by the authors.
> >
> > > Intuitively, for a heterophilic graph, where the number of direct connections between similar nodes is less than the dissimilar ones, the additional paths are likely to reduce the average shortest path length for similar nodes comparatively more than the dissimilar nodes, thus increasing the homophily. Similarly, in the homophilic graph, where the number of direct connections between dissimilar nodes is less than the similar ones, it brings vertices with dissimilar labels closer to each other than the similar ones, thus decreasing the homophily.
> >
> > This response agrees with my intuition and Table 3 that edges with "neutral" homophily (on average) are added. The fact that it increases homophily for highly heterophilic graphs does not help to understand why the performance is expected to increase. It is known from several previous works on homophily that GNNs work well on strongly homophilic graphs and often also work well on strongly heterophilic graphs. The biggest challenge for GNNs is graphs that are neutral in terms of homophily.
> >
> > > We thank you for pointing out this. We have corrected the typo in the revised paper.
> >
> > Regarding Equation (1), I  still do not understand why it is a mathematical formulation of the approach. It is written that (1) describes "engineering technique outlined by the following process",  but I do not see any process in these formulas. There is a model notation followed by the implication sign (note that this symbol has its mathematical meaning, but I do not know why it is used in this formula); after the implication sign, there are two equivalent graphs. So, I would not say that it is a mathematical formulation of any process.

---

> ### Author Response · Authors · 2024-12-01
> **Response to Reviewer bR69 - Part 4**
>
> Dear Reviewer bR69,
>
> Thanks for appreciating our detailed response to your questions. We take this opportunity to clarify your concerns further.
>
> ***
> **Q1:** In general, I do not think that the fact that many papers use some potentially flawed datasets justifies their usefulness. For instance, the original Squirrel and Chameleon datasets are also often used in the papers, while they have significant train-test leakage. The same applies to relying on accuracy for imbalanced datasets.
>
> **Q2:** Yes, I understand that the filtered versions are used, but, as I understand, Platonov et al. do not guarantee that the filtration produces meaningful datasets and thus these datasets are not included in the list of five datasets proposed by the authors.
>
> **Response to Q1 and Q2:** Thanks for acknowledging that we used the filtered versions of Squirrel and Chameleon datasets, which are devoid of any train-test leakage. Platonov et al. released these corrected datasets which are meaningful as they resolved the leakage issue. Further, regarding accuracy, we reiterate that to the best of our knowledge, a long list of existing and latest literature employs accuracy as the metric, which forms the basis of our usage as well.
>
> ***
>
> **Q3:** This response agrees with my intuition and Table 3 that edges with "neutral" homophily (on average) are added. The fact that it increases homophily for highly heterophilic graphs does not help to understand why the performance is expected to increase. It is known from several previous works on homophily that GNNs work well on strongly homophilic graphs and often also work well on strongly heterophilic graphs. The biggest challenge for GNNs is graphs that are neutral in terms of homophily.
>
> **Response:** We have answered this question in detail in lines 348-354 with appropriate citations. Here, we show it again for your perusal.
>
> *“At present, two different lines of thought prevail in the GRL field. One set of work discusses the
> performance of GNNs regardless of the homophily levels (Luan et al., 2023), or the idea of good
> homophily and bad homophily (Ma et al., 2022), or the heterophily not always being harmful to
> GNN’s performance (Luan et al., 2022). The other set of work shows that GNN’s performance is
> indeed proportional to the homophily (Rossi et al., 2024; Liu et al., 2024; Xu et al., 2023; Suresh
> et al., 2021). DirGNN (Rossi et al., 2024) showed that treating graphs as directed improves learning on heterophilic graphs and attributed it to the increase in homophily.”*
>
> ***
>
> **Q4:** Regarding Equation (1), I  still do not understand why it is a mathematical formulation of the approach. It is written that (1) describes "engineering technique outlined by the following process", but I do not see any process in these formulas. There is a model notation followed by the implication sign (note that this symbol has its mathematical meaning, but I do not know why it is used in this formula); after the implication sign, there are two equivalent graphs. So, I would not say that it is a mathematical formulation of any process.
>
> **Response:** We acknowledge your concern about the wording of this statement and the mathematical notation of not probably being in sync. We will rephrase it accordingly for better suitability in case you find other aspects of our research suitable for acceptance.
>
> ***
> **Q5:** Thank you for the experiments on a new dataset, I think that it is important for validating the claims of the paper. I have one additional question regarding the results. Could you please explain the difference in performance between your results and the results in Lim et al. (2021)? This applies to the arXiv-year and twitch-gamers datasets. In particular, the results for the LINKX reported by Lim et al. (2021) are 56 for arXiv-year and 66 for twitch-gamers, which is larger than all the values reported for these datasets in the current paper, both with or without SH. I see similar differences for some other models, so I wonder whether it is explained by differences in data preprocessing (e.g. splits) or training process.
>
> **Response:** We appreciate you for this observation. The observed deltas are related to and explained by the differences in random seeds used for data splits.

---

### Official Review · Reviewer_e6x4 · 2024-11-06

**Soundness:** 2
**Presentation:** 2
**Contribution:** 2
**Rating:** 5
**Confidence:** 4

**Summary:**

The paper proposes Spectral Highways, a technique that enhances the performance of Graph Neural Networks on heterophilic graphs with additional nodes and connections forming highways over the original graph.

**Strengths:**

1. The story is interesting.
2. It tested on rich datasets.

**Weaknesses:**

1. The addition of nodes and connections is a common approach in graph augmentation. For example, methods such as [1] use similar upsampling approaches. To better showcase the novelty, I suggest that the authors emphasize these unique aspects more explicitly and explore how these modifications lead to improvements over [1] and similar methods.
2. While enriching the graph can enhance performance, it may also introduce considerable computational overhead, especially on large-scale datasets. I recommend that the authors include a detailed analysis of time and space complexity. Specifically, it would be beneficial to compare the construction time of Spectral Highways relative to the original graph size, as well as its impact on downstream GNN runtime.
3. Since datasets are already introduced in Section 2.1, Section 2.2 could be streamlined to focus primarily on algorithmic contributions without redundancies. In table 2, The analysis could be expanded to include comparisons that consider performance gains across dataset characteristics, such as homophily levels, graph sizes, and class distributions. This additional detail would make the results more informative and enable better comparisons across datasets.

[1] Azabou, Mehdi, et al. "Half-Hop: A graph upsampling approach for slowing down message passing." International Conference on Machine Learning. PMLR, 2023.

**Questions:**

See Weakness.

---

> ### Author Response · Authors · 2024-11-25
> **Response to Reviewer e6x4**
>
> Dear Reviewer e6x4,
>
> Most importantly, we thank you for finding our work interesting and also for approving the rich coverage of datasets.
>
> Below, we tried to incorporate your comments in the best way possible.
> ***
> **W1**: The addition of nodes and connections is a common approach in graph augmentation. For example, methods such as [1] use similar upsampling approaches. To better showcase the novelty, I suggest that the authors emphasize these unique aspects more explicitly and explore how these modifications lead to improvements over [1] and similar methods.
>
> **Response**: We have now incorporated the results of the suggested augmentation method, namely HalfHop, in Table 2. Also, we incorporated one additional and very recent work, IPRMPNN, as pointed out by another reviewer. We request that you kindly have a look at our revised manuscript. We observe that our proposed approach augmentation results in much better results than the augmentation approach of [1] (HalfHop) on all datasets. The other method, IPRMPNN, results are better than HalfHop but still lower than our proposed method on all the datasets.
> ***
> **W2**: While enriching the graph can enhance performance, it may also introduce considerable computational overhead, especially on large-scale datasets. I recommend that the authors include a detailed analysis of time and space complexity. Specifically, it would be beneficial to compare the construction time of Spectral Highways relative to the original graph size, as well as its impact on downstream GNN runtime.
>
> **Response**: Thanks for suggesting interesting experiments. We have included the suggested analysis in Appendix sections C and D in the revised manuscript.
> ***
> **W3**: Since datasets are already introduced in Section 2.1, Section 2.2 could be streamlined to focus primarily on algorithmic contributions without redundancies. In table 2, The analysis could be expanded to include comparisons that consider performance gains across dataset characteristics, such as homophily levels, graph sizes, and class distributions. This additional detail would make the results more informative and enable better comparisons across datasets.
>
> **Response**: Section 2 covers the related work. We utilised Section 2.2 to highlight the generic and specialised heterophilic GNNs that exist in the literature. We have utilised the introduction to focus on algorithmic contributions. We sincerely hope you do not consider our attempt to cover existing works in detail as a weakness. Unfortunately, Table 2 does not provide us with much space to include all the comparisons in it. Hence, we have discussed these comparisons in the Analysis and Discussion Section 6 in Tables 2, Table 3, Table 4 and Table 5.
> ***
> To summarise, your feedback was of utmost value, and we tried to accommodate it in the best way possible.

---

### Official Review · Reviewer_bUwU · 2024-11-10

**Soundness:** 2
**Presentation:** 3
**Contribution:** 2
**Rating:** 3
**Confidence:** 4

**Summary:**

This paper presents a graph data augmentation method called Spectral Highway (SH). It involves dividing existing nodes into clusters using spectral clustering and then adding a layer of nodes on top of them. Each new node corresponds to a spectral cluster in the original graph. The new nodes form a fully connected subgraph, and they are connected to the original nodes belonging to their respective clusters in a principled way.

**Experiments.**
The authors tested the improvement in node classification performance on several classic *small-scale* datasets, showing *significant* results on ChameleonF and SquirrelF. They also compared SH with several rewiring and data-centric methods, demonstrating the advantages of SH as a graph augmentation method.

**Analysis.**
The authors mainly considered the impact of SH on homophily *intuitively and through experiments*, as well as the effect on intra-class and inter-class average shortest path lengths.

**Strengths:**

1. The method and illustrations in the paper are clear.
2. The approach is very simple, but it shows significant results on ChameleonF and SquirrelF.
3. The paper organizes existing motivations for heterophilic graphs and addresses them in the analysis section.

**Weaknesses:**

1. The algorithm is difficult to run on larger datasets. The paper emphasizes the impact of random seeds, which is reflected in the large variance of results. This is largely due to the small size of the datasets involved (Cornell, Texas, Wisconsin). Because of this, the authors should conduct experiments on larger datasets. However, the algorithm depends on spectral clustering, which seems to prevent it from scaling to larger datasets.

2. Few rewiring, data-centric/graph-augmentation methods are compared in the paper.

3. The paper empirically discusses SH’s impact on homophily across various metrics. However, aside from the decrease in Aggregated Homophily on many datasets, the observed improvements are *not significant* in terms of numerical values.

**Questions:**

Please check weaknesses.

---

> ### Author Response · Authors · 2024-11-25
> **Response to Reviewer bUwU**
>
> Dear Reviewer bUwU,
>
> Firstly, we sincerely thank you for appreciating the presentation of our work, including the method, illustrations, motivation, analysis and paper organisation. It means a lot.
>
> Here, we would like to clarify the pointed weaknesses to the best of our ability.
> ***
> **W1**: The algorithm is difficult to run on larger datasets. The paper emphasizes the impact of random seeds, which is reflected in the large variance of results. This is largely due to the small size of the datasets involved (Cornell, Texas, Wisconsin). Because of this, the authors should conduct experiments on larger datasets. However, the algorithm depends on spectral clustering, which seems to prevent it from scaling to larger datasets.
>
> **Response**:  For small datasets, the variance is higher. Yes, as observed we uncovered the high sensitivity of several recently proposed GNNs to the random seed used. However, the large variance of results is not solely related to the size of the datasets. From the results in Table 1, we can observe huge variance for larger datasets like arXiv, too. For example, we see very high standard deviations of 18.39, 12.85, 15.19, and 15.59 for AeroGNN, PMLPGCN, PMLPAPPNP, and UniFilter algorithms on arXiv, respectively. As discussed in the paper, we do not observe this large variance for the rest of the algorithms. We already have experiments on one large dataset namely Arxiv. We have started experiments on another large dataset and we should be able to report results on that soon.
>
> Yes, the spectral clustering is a costly algorithm. However we observe the best results with using it. We did not use efficient implementations of the spectral clustering algorithm. The clustering is a one time operation in the proposed pipeline. Also, one can use existing parallel and scalable implementations of Spectral clustering to handle very large datasets. Furthermore, we have now included the time and space analysis in Appendix sections C and D.
>
> ***
> **W2**: Few rewiring, data-centric/graph-augmentation methods are compared in the paper.
>
> **Response**: We appreciate this concern. We have added two very recent data-centric/graph-augmentation methods as suggested by the other two reviewers. Please see the inclusion in Table 2 in updated paper.
> ***
> **W3**: The paper empirically discusses SH’s impact on homophily across various metrics. However, aside from the decrease in Aggregated Homophily on many datasets, the observed improvements are not significant in terms of numerical values.
>
> **Response**: We here take the opportunity to clarify this perceived weakness. We have seen improvements in adjusted homophily scores of even more than 100% across heterophilic datasets as shown in Table 3. Since the original homophily scores for heterophilic datasets are already very low, hence even after achieving more than a 100% increment, the absolute values of new scores still lie in a low range. Further, we would like to reinforce the fact that our algorithm injects homophily into heterophilic graphs and does not change the heterophilic graph into a homophilic graph. This can indeed be a good goal to explore in future.
>
> ***
> To conclude, we thank you for highlighting your concerns, and we have constructively received your feedback.

---

> > ### Author Response · Authors · 2024-11-29
> > **Response to Reviewer bUwU - Part 2**
> >
> > Dear Reviewer bUwU,
> >
> > Please find below the results for a **large** heterophilic dataset, Twitch-Gamers, containing 1,68,114 nodes and 67,97,557 edges.
> >
> > ***
> >
> > | Model | GraphSAGE | GAT | APPNP | LINKX | GATv2 | PGNN | DAGNN | DirSAGE | PMLPGCN | PMLPAPPNP | UniFilter |
> > |-------------|:-------------:|:------:|:-------------:|:-----:|:-------------:|:-----:|:-----:|:-------------:|:-----:|:-------------:|:-----:|
> > | **G**   |59.26 ± 0.63 |52.04 ± 6.84 |59.66 ± 0.57 |62.62 ± 1.62 |50.60 ± 7.12 |60.06 ± 1.08 |58.49 ± 4.16 |60.36 ± 0.76 |48.68 ± 10.79 |50.47± 8.32 |57.89± 5.44|
> > | **SH** |60.54 ± 0.27 |54.95 ± 5.76 |60.77 ± 0.63 |65.51 ± 1.49 |52.18 ± 6.83 |60.99 ± 1.21 |59.64 ± 1.96 |62.55 ± 0.68 |52.78 ± 6.10 |51.74 ± 9.65|59.04 ± 3.69 |

---

> > > ### Comment · Reviewer_bUwU · 2024-12-01
> > > **Response to Reviewer Authors**
> > >
> > > Thanks for the reply.
> > >
> > > ---
> > >
> > > **In response for W1**：
> > >  > Furthermore, we have now included the time and space analysis in Appendix sections C and D.
> > >
> > >  In Appendix sections C and D, you provide running time for PageRank and DivRank, instead of spectral clustering, right?
> > >
> > >  Anyway, I appreciate that you add an experiment on Twitch-gamer.
> > >
> > > ---
> > >
> > > **W4**: According to Table 7, SH only works on heterophilic graphs. But at the same time, as shown in Table 1&2, SH is not compared to stronger baselines in heterophilic graphs, e.g., Table.1 in [1].
> > >
> > >
> > > ---
> > > **Q**: The authors aim at improving homophily on heterophilic graphs. I am not sure whether it's right -- for example, [2] shows that not all cases of heterophily are harmful.
> > >
> > > [1] Guo et al. Graph neural networks with learnable and optimal polynomial bases.
> > >
> > > [2] Luan et al. Is heterophily a real nightmare for graph neural networks to do node classification?

---

> ### Author Response · Authors · 2024-12-01
> **Response to Reviewer bUwU - Part 3**
>
> Dear Reviewer bUwU,
>
> Firstly, we thank you for appreciating our efforts. It means a lot. Below, we clarify your further two questions and a newly pointed weakness.
>
> ***
> **Q1:** In Appendix sections C and D, you provide running time for PageRank and DivRank, instead of spectral clustering, right?
>
> **Response:** As mentioned in lines 942-945, we “Analyse the construction time of augmented graph which **includes** the time taken for (i) **Spectral Clustering**, (ii) running ranking algorithm, (iii) forming connections between spectral nodes and the original graph and also amongst themselves, and (iv) label and feature assignment of spectral nodes utilising likelihood estimation.”
>
> ***
> **W4:** According to Table 7, SH only works on heterophilic graphs. But at the same time, as shown in Table 1&2, SH is not compared to stronger baselines in heterophilic graphs, e.g., Table.1 in [1].
>
> **Response:** We would have dearly liked to incorporate this additional baseline had it been referred earlier. As mentioned in the manuscript, DirSAGE is the latest and strongest performing baseline for heterophilic graphs. We have already added two baselines as suggested by the other two reviewers in Table 2 and showed that the merit of our proposed algorithm still prevails. Further to highlight, we have conducted thorough extensive experimentation and comparison showcasing our performance against 16 baseline GNNs in Table1 and 9 baseline rewiring models in Table2.
>
> ***
> **Q2:** The authors aim at improving homophily on heterophilic graphs. I am not sure whether it's right -- for example, [2] shows that not all cases of heterophily are harmful.
>
> **Response:** We would like to highlight that the referenced work [2] is a rejected paper from ICLR 2022, and **we have already discussed its accepted version [3]** from NeurIPS 2022 in our manuscript in **Section 5 and Table 2**. Moreover, we also uncovered the high sensitivity of this referred work w.r.t. the random seed used for data splitting.
> ***
>
> **[3]** Sitao Luan, Chenqing Hua, Qincheng Lu, Jiaqi Zhu, Mingde Zhao, Shuyuan Zhang, Xiao-Wen Chang, and Doina Precup. Revisiting heterophily for graph neural networks.
>
> ***
> We believe that we have sincerely discussed these points in our original manuscript and have further addressed them in our revised version.

---

### Public Comment · ~Arshdeep_Singla1 · 2024-11-22
**A few queries**

Hi, I stumbled upon this work while looking for specialized algos for heterophilic graphs. I work with financial data on asset management here at Berkeley and continuously look for the latest models for my leverage. I thoroughly enjoyed reading this paper smoothly, especially the sensitivity of many of the latest GNNs to random seed, and the uncovering of shortest path length phenomena and its relationship to the various homophily measures and model performance. It was pretty insightful that I could thoughtfully perturb my existing data using spectral clustering and probabilistic modeling to achieve better predictions. I currently use state of the art 'Dirsage' model for my data, and you have shown significant improvements over it through extensive detailed experimentation, which intrigued me. I have a few queries.

Since this perturbation is an augmentation methodology, can you please let me know the average number of additional nodes and edges it introduces?

Also, can you tell me which ranking algorithm you prefer to choose the nodes for spectral connections?

And do you plan to release your code in the near future? I would greatly appreciate that.

Thanks in advance.

---

> ### Author Response · Authors · 2024-11-25
> **Response to a few queries**
>
> Dear Arshdeep:
>
> Firstly, we would like to express our sincere gratitude for finding our research interesting and useful in your line of work. We also thank you for highlighting the merits of our research.
> ***
> **Q1**: Since this perturbation is an augmentation methodology, can you please let me know the average number of additional nodes and edges it introduces?
>
> **Response**: Thanks for asking this query. We have added a detailed response in Appendix Section D of the revised manuscript. Please have a look at it.
>
> ***
> **Q2**: Also, can you tell me which ranking algorithm you prefer to choose the nodes for spectral connections?
>
> **Response**: We worked diligently upon it and have incorporated the analysis and a detailed response in Appendix Section C of the revised manuscript.
>
> ***
> **Q3**: And do you plan to release your code in the near future? I would greatly appreciate that.
>
> **Response**: We will release our code publicly upon acceptance.

---

> > ### Public Comment · ~Arshdeep_Singla1 · 2024-11-25
> >
> > I sincerely thank you for caring enough to reply to my queries in detail and perform new experiments to publish results and derive inferences. After going through the results of time and space analysis, I believe it will go well with my use cases. Best of luck with your paper acceptance!

---

### Public Comment · ~Samyak_Jain3 · 2024-11-25
**Impact of spectral clustering**

Dear Authors,\
I am quite fascinated by your approach of employing traditional ML concepts of Spectral clustering for graph node prediction in a heterophilic setting. It is extremely interesting that you play with the notion of path lengths to bring dissimilar vertices closer to inject homophily. I see its applicability in many real-life industrial challenges that we face. Though I really liked your analysis and discussion segment, however, I couldn’t find any discussion where we see the impact of spectral clustering on downstream prediction. If you plan to rebut, can you please show these results if possible? Also, for the purpose of utilizing your work do you also plan to open source the code?

---

> ### Author Response · Authors · 2024-11-27
> **Response to Impact of spectral clustering**
>
> Dear Samyak,
>
> Firstly, we would like to thank you for finding our research fascinating and appreciating our notion of path length perturbation. We are glad that you carefully went through our Analysis and Discussion section and found it interesting.
>
> **Q1**: Though I really liked your analysis and discussion segment, however, I couldn’t find any discussion where we see the impact of spectral clustering on downstream prediction. If you plan to rebut, can you please show these results if possible?
>
> **Response**: We take your feedback very seriously and have performed an ablation study to see the impact of Spectral Clustering on downstream prediction. **Additionally**, we have done one more ablation study to gauge the importance of our design of probabilistic modelling for labelling spectral nodes. We refer you to Appendix Section B in the latest revised manuscript for this purpose.
>
> ***
> **Q2**:  Also, for the purpose of utilizing your work do you also plan to open source the code?
>
> **Response**: We will release our code publicly upon acceptance.

---

> > ### Public Comment · ~Samyak_Jain3 · 2024-11-28
> >
> > Thanks a lot for addressing my queries promptly!

---

### Author Response · Authors · 2024-11-29
**Final and revised manuscript has been uploaded**

Dear Reviewers,

We have tried our best to address your concerns and revised the paper. We have uploaded the final version of our paper. Compared to the previous version, we made the following major changes:


**Addition of more baselines:** We have added 2 additional baselines as suggested by Reviewers bUwU, e6x4 and bR69. We discussed and compared the results in the revised manuscript.

**Conducting ablation studies:** We have conducted 3 ablation studies following the request of Reviewer dhKQ and the public commentator. The studies highlight the importance of each step of our proposed algorithm.

**Revising Introduction:** We have revised the introduction following the suggestion from Reviewer dhKQ to make it more coherent.

**Adding large dataset:** Following the suggestion of Reviewers bUwU and bR69, we ran experiments on one large dataset and showed results in the discussion forum.

**Time and Space Analysis:** We have added the relevant sections covering the time and space aspects of our proposed method in the revised manuscript. We thank Reviewer e6x4 and the public commentator for this feedback.

**Minor fixes:** We have corrected the typos and the suggested minor fixes as suggested by Reviewer bR69.


We will be grateful if you can kindly take a look at our revised manuscript and the rebuttal responses. **We have tried our best to produce a lucid and thorough rebuttal**. We also fully understand that you probably need to prioritise your own submissions at this moment. We wish all of you good luck at this extended discussion stage.

Sincere Regards.

---

### Meta-Review · Area_Chair_ahs6 · 2024-12-20

**Metareview:**

The paper introduces a graph augmentation method called Spectral Highway (SH) to improve the performance of Graph Neural Networks (GNNs) on heterophilic graphs by adding supernodes that represent spectral clusters. The method is evaluated on several small-scale datasets, showing promising results, particularly on ChameleonF and SquirrelF. The paper also explores the impact of SH on homophily and node classification performance, though its broader applicability and the significance of the improvements are not fully explored.

While the proposed model shows promising results, the reviewers have identified several weaknesses that need to be addressed:

1. The paper lacks a clear theoretical explanation for why the method is effective on heterophilic graphs, with empirical results sometimes showing mixed or insignificant improvements.
2. The method struggles to scale to larger datasets due to its reliance on spectral clustering, which limits its applicability to bigger graphs.
3.  The paper only compares SH to a few rewiring and graph augmentation methods, missing a broader range of relevant approaches in the literature.

Based on these weaknesses, we recommend rejecting this paper. We hope this feedback helps the authors improve their paper.

**Additional Comments On Reviewer Discussion:**

The authors have revised the paper by adding more baselines, conducting three ablation studies, and incorporating experiments on a larger dataset as suggested by reviewers. They also revised the introduction for clarity, included time and space complexity analyses, and addressed minor issues like typos and formatting errors.

However, the reviewers’ concerns regarding insufficient theoretical justification and scalability issues remain unaddressed during the rebuttal and discussion phase. As such, I recommend rejection based on the persistent issues highlighted in the reviewers’ feedback.

---

### Decision · Program_Chairs · 2025-01-22

Reject